

# Research progress of nano drug delivery systems in the anti-tumor treatment of traditional Chinese medicine monomers

Bocui Song[1], Li Shuang[1], Shuang Zhang[2], Chunyu Tong[1], Qian Chen[1], Yuqi Li[1], Meihan Hao[1], Wenqi Niu[1] and Cheng-Hao Jin[1,3]

[1] College of Life Science and Technology, Heilongjiang Bayi Agricultural University, Daqing, Heilongjiang, China
[2] Comprehensive Service Center, Yongji Economic Development Zone, Jilin, Jilin, China
[3] College of Life Science and Technology, College of Life Science and Technology, Daqing, Heilongjiang, China

Corresponding authors
Bocui Song, songbocui66@163.com
Cheng-Hao Jin,
jinchenghao3727@byau.edu.cn

## ABSTRACT

Tumors pose a serious threat to global public health and are usually treated from two aspects: tumor cells and tumor microenvironment. Compared with traditional chemotherapy drugs, traditional Chinese medicine (TCM) monomers have advantages in tumor treatment, such as multiple targets, multiple levels and synergistic intervention. However, most TCM active ingredients have disadvantages such as poor water solubility and stability, which restrict their clinical application. Nano drug delivery systems have the functions of improving the bioavailability of TCM anti-tumor active ingredients, enhancing tissue targeting, achieving controlled drug release, and inhibiting tumor multidrug resistance. Compared with free monomers, they have higher therapeutic effects and fewer side effects. This article summarizes five commonly used anti-tumor TCM monomer nanocarriers, including lipid nanomaterials, exosomes, polymer micelles, carbon nanotubes, and dendrimers, and explains their anti-tumor mechanisms after combining with TCM, such as inhibiting tumor cell proliferation and metastasis, regulating tumor microenvironment, *etc*. At the same time, the potential of nano drug delivery systems combined with radiotherapy and immunotherapy is discussed, as well as the current problems of potential toxicity, long-term stability, and complex amplification process, as well as future development directions, aiming to provide a reference for promoting the clinical application of nano drug delivery systems for TCM anti-tumor active ingredients.

## INTRODUCTION

With the aging of China's population, the incidence and mortality of tumors have increased significantly. The number of new tumor patients and deaths accounts for about a quarter of the world. As a major disease threatening human health, tumors have become a global public health problem and a major health burden for the human body. The treatment methods for tumors include radiotherapy (*Petroni et al., 2022*), chemotherapy

(*Liang et al., 2022*), targeted therapy (*Min & Lee, 2022*), immunotherapy (*Ribas & Wolchok, 2018*), endocrine therapy (*Gardner, Park & Allen, 2024*) and other methods. Among them, chemotherapy is the most commonly used method. The therapeutic mechanism of chemotherapy includes affecting the chemical structure of DNA, inhibiting nucleic acid synthesis, and interfering with mitotic micro-protein synthesis (*Dickens & Ahmed, 2021*). The chemotherapy drugs currently used include a series of chemical drugs such as doxorubicin, gemcitabine, methotrexate, docetaxel, *etc.* (*Wang et al., 2016*). However, due to the fact that there is no fundamental difference in the metabolism of tumor cells and normal cells, most chemical drugs have poor targeting, are prone to adverse reactions such as bone marrow suppression, liver and kidney dysfunction, and are expensive.

Compared with conventional anti-tumor drugs, natural drugs have the characteristics of multiple components, multiple pathways and multiple targets. In fact, the application of traditional Chinese medicine (TCM) to treat tumors has a long history. TCM treatment of malignant tumors emphasizes "holistic concept" and "living with tumors". The treatment goal is not only limited to killing cancer cells and shrinking tumors, but also to improve the quality of life of patients and reduce toxic side effects. In the ancient book "Puji Fang", tumors are recorded as "accumulation of lumps and masses". In "Shennong's Herbal Classic", 71 anti-tumor TCM monomers are recorded, and in "Compendium of Materia Medica", 214 drugs with therapeutic effects on tumors are recorded (*Hou & Zhou, 2015*; *Li et al., 2025*). TCM monomers can achieve significant tumor inhibition effects through multiple targets and multiple pathways. At present, a total of 61 TCM monomers with anti-tumor effects have been extracted from TCM, including one steroid (bufalin), one organic acid (betulinic acid), three phenylpropanoids (verbascoside, *etc.*), four quinones (shikonin, *etc.*), five phenols (emodin, *etc.*), five saponins (ginsenoside Rg3, *etc.*), nine terpenes (paclitaxel, *etc.*), 15 alkaloids (berberine, *etc.*), and 17 flavonoids (rutin, *etc.*) (*Qian et al., 2020*; *Cai, 2023*; *Jia et al., 2020*; *Ma et al., 2020*; *Deng et al., 2023a*; *Wang et al., 2024a*; *Scribano et al., 2021*; *Sun et al., 2022*; *Zhu et al., 2021*). The anti-tumor mechanism is shown in Table 1. Although tumor microenvironment (TME) have obvious anti-tumor effects and can exert their effects through mechanisms and pathways such as inhibiting tumor cell proliferation, promoting tumor cell apoptosis and regulating the TME, their own limitations hinder their clinical transformation. Many TCM monomers need to reach a certain blood drug concentration in the body to exert their effects, but most of the poorly soluble components have poor solubility and stability, resulting in low bioavailability, and are easily oxidized, hydrolyzed or isomerized, resulting in unsatisfactory tumor target treatment. There are also some active ingredients of TCM with large relative molecular weight and weak lipophilicity, which have low transport efficiency and poor permeability. Therefore, after entering the body, the monomers of TCM have the disadvantages of fast component clearance and short blood circulation cycle, making it difficult to penetrate the biological membrane to exert their efficacy.

In the 1960s, chemists proposed to apply nanomaterials to the field of biopharmaceuticals and prepare a nano drug delivery system (DDS) to improve the therapeutic effect of drugs (*Zhang et al., 2024*). Loading TCM monomers into

**Table 1 Mechanism of anti-tumor action of TCM monomers.**

| Classify | TCM monomers | Chemical formula of TCM monomers | Source | Experimental model | Changes in tumor cells and tumors | Antitumor mechanisms | References |
|---|---|---|---|---|---|---|---|
| Steroid | Bufalin |  | *Bufo gargarizans* | Human gallbladder cancer cell lines GBC-SD and SGC-996 cells and Nude mice model with subcutaneous GBC-SD cell xenografts | Inhibited cell proliferation, induced apoptosis and cell cycle arrest, reduced tumor volume by 40% | Bufalin inhibits c-Met-mediated MEK/ERK and PI3K/AKT signaling pathways, suppressing tumor cell invasion, migration, and self-renewal of cancer stem cells. Bufalin (2 mg/kg) significantly reduced tumor volume by about 40% and effectively inhibited the growth of gallbladder cancer cells. | (*Qian et al., 2020*) |
| Organic acids | Betulinic acid |  | *Betula platyphylla* Suk | GBC-SD and SGC-996 cells, Nude mice model with subcutaneous GBC-SD cell xenografts | Inhibited cell migration and clonogenic ability | Derivative Q-4 of betulinic acid induces apoptosis, autophagy, and cell cycle arrest, producing reactive oxygen species. Q-4 exerts anti-tumor effects by inhibiting the PI3K/AKT/mTOR signaling pathway and reversing trastuzumab resistance. | (*Cai, 2023*) |
| Phenylpropanoid | Verbascoside |  | *Cistanche deserticola* | MCF-7, Hela, and MDA-MB-231 cells | Significantly inhibited proliferation, migration, and invasion of GBM cells, promoted apoptosis and autophagy. Tumor volume was significantly reduced compared to the control group. | Verbascoside significantly inhibits the proliferation, invasion, and migration of colorectal cancer cells by inhibiting the Ras-related C3 botulinum toxin substrate 1 (Rac-1), hypoxia-inducible factor-1α (HIF-1α), and zinc finger E-box binding homeobox 1 (Zeb-1) signaling pathways. | (*Jia et al., 2020*) |
| Quinones | Shikonin |  | *Lithospermum erythrorhizon* | Hs683 cells | Induced and promoted apoptosis | Promoted apoptosis by activating Caspase-3 and upregulating PERK/CHOP protein levels. | (*Ma et al., 2020*) |

(Continued)

| Classify | TCM monomers | Chemical formula of TCM monomers | Source | Experimental model | Changes in tumor cells and tumors | Antitumor mechanisms | References |
|---|---|---|---|---|---|---|---|
| Phenols | Emodin |  | *Reynoutria japonica* | HaCaT cells | Regulated cell cycle, promoted apoptosis | Insulin growth factor receptor (IGF1R) is the key target of emodin in the treatment of gastric cancer. Emodin induces apoptosis of gastric cancer cells and inhibits proliferation by inhibiting the IGF1R/PDK1 signaling pathway. | (*Deng et al., 2023a*) |
| Saponin | Ginsenoside Rg3 |  | *Panax ginseng* | NSCLC cell lines H1975, H1299, A549, HCC827 and LLC cell xenograft model | Inhibited tumor growth, reduced tumor volume | Rg3 inhibited the glycosylation of PD-L1 by suppressing the EGFR signaling pathway, thereby enhancing T cell-mediated anti-tumor immune responses. *In vivo* experiments showed that Rg3 dose-dependently increased the concentration of IL-2, IFN-γ, and TNF-α in mouse serum, while promoting the activation of CD8+ T cells in the tumor microenvironment, as evidenced by increased expression levels of perforin and granzyme B. | (*Wang et al., 2024a*) |
| Terpenes | Paclitaxel |  | *Taxus brevifolia* | MDA-MB-231, Cal51, MCF10A and LLC cell xenograft model in C57/BL6J mice | Enhanced cytotoxicity, reduced tumor volume | Increasing the duration of multipolar spindle formation significantly enhanced the cytotoxicity of paclitaxel. Inhibiting Mps1 or upregulating Mad1 reduced the efficacy of paclitaxel. Mad1-upregulated MDA-MB-231 cells exhibited lower sensitivity to paclitaxel, with less tumor reduction compared to the control group. Paclitaxel exerts anti-tumor effects by inducing multipolar spindle formation and chromosomal instability. | (*Scribano et al., 2021*) |

| Classify | TCM monomers | Chemical formula of TCM monomers | Source | Experimental model | Changes in tumor cells and tumors | Antitumor mechanisms | References |
|---|---|---|---|---|---|---|---|
| Alkaloids | Berberine |  | Coptis chinensisFranch | HT-29 colorectal cancer cells and CRC mouse model | Inhibited proliferation, migration, invasion, and clonogenic ability of HT-29 cells, induced apoptosis | Berberine concentration- and time-dependently reduced the levels of anti-apoptotic protein Bcl-2, upregulated pro-apoptotic protein Bax expression, and activated Caspase-9 and Caspase-3. By inhibiting the abnormal activation of the Hedgehog signaling pathway, it reduced the expression of SHH, Ptch1, SMO, Gli1, and c-Myc, while enhancing the expression of SUFU, thereby suppressing the malignant phenotype of CRC cells. *In vivo* experiments showed that berberine significantly alleviated the pathological features of AOM/DSS-induced CRC mice, increased survival rates, relieved colon shortening, and reduced the activity of the Hedgehog signaling pathway. | (*Sun et al., 2022*) |
| Flavonoids | Puerarin |  | Rutagravolelensl | PANC-1 and PATU-8988T cell lines, nude mice model with subcutaneous xenografts | Inhibited cell proliferation, induced mitochondria-mediated apoptosis, reduced tumor mass and volume | Bax/Bcl-2 imbalance, puerarin concentration- and time-dependently inhibited epithelial-mesenchymal transition (EMT) to suppress cell migration and invasion. Puerarin reduced Ki67 and c-Myc expression, upregulated Cleaved caspase-8 and Bax, downregulated Bcl-2 expression, while inhibiting the expression of EMT-related protein α-SMA and increasing E-cadherin expression. | (*Zhu et al., 2021*) |

nanomaterials to make nano drug delivery systems is an innovative strategy for developing TCM monomers to treat tumors. After separating and purifying TCM monomers from natural plant TCMs, they are optimized with nanotechnology, which can not only maximize the efficacy of the drugs, but also avoid the complex mechanisms between the multiple components of TCM. In the nano TCM monomer delivery system, the TCM monomer plays a therapeutic role, and nanomaterials can enhance the pharmacological effects of TCM monomers in different ways (*Zheng et al., 2022*). For example, nanocarriers have a good ability to dissolve hydrophobic drugs, increase drug solubility, accelerate drug penetration through biological barriers, and improve bioavailability; by connecting specific antibodies, ligands or certain stimuli-sensitive systems to nanocarriers, the release of TCM monomers can be controlled to achieve targeted delivery of anti-tumor drugs, solve the non-targeting problem of conventional chemotherapy drugs, improve the targeting effect of drugs, and inhibit tumor multidrug resistance of TCM monomers. The active ingredients of TCM have the characteristics of multi-target effects. After being coated with nanomaterials, they can enhance the anti-tumor effect of drugs in terms of affecting tumor cells and regulating TME.

This study searched PubMed, Web of Science, Science Direct, and China National Knowledge Infrastructure (CNKI) databases within 5 years. The inclusion criteria were articles on the antitumor effects of TCM monomers and nanoparticles. The exclusion criteria were articles involving traditional Chinese medicines and nanoformulations, we did not refer to articles using nano-TCMs instead of nano-TCM monomers, and articles that did not discuss the use of nano-TCM drug delivery systems to treat other diseases. After applying the inclusion and exclusion criteria, 108 articles were selected from 1,380 articles for systematic review to comprehensively summarize the therapeutic effects of nanomaterials and TCM combined with each other on tumor diseases.

## SURVEY METHODOLOGY

After determining the title of the article, we searched PubMed, Web of Science, Science Direct and China National Knowledge Infrastructure (CNKI) databases with the keywords "nanotechnology", "traditional Chinese medicine monomers" and "anti-tumor drugs". For the search results, we read the article titles and abstracts as well as selected articles matching the theme of anti-tumor traditional Chinese medicine. In order to maintain the novelty of the article, we mainly cited articles within 5 years. For the three main topics in the article, we added "nano drug delivery system", "anti-tumor mechanism" and "type of nanomaterials" to the original keywords. We read the selected articles, searched again, and screened again. A total of 108 articles were selected for citation in the order of the most consistent with the research content, the most recent publication, and the highest impact factor to illustrate the anti-tumor effect and mechanism of the combination of nanomaterials and traditional Chinese medicine monomers.

## PRINCIPLES OF NANODRUG DESIGN

Cancer poses a serious threat to global public health. In the context of China's aging population, its morbidity and mortality rates have increased significantly. Given the grim

status of cancer treatment, chemotherapy, as a common method, has poor targeting, easy to induce adverse reactions such as bone marrow suppression and liver and kidney dysfunction due to the small metabolic differences between tumors and normal cells, and is expensive, which limits the therapeutic effect. TCM monomers have the advantages of multi-target effects, low toxicity and great synergistic potential in tumor treatment (*Wei et al., 2023*). However, most active ingredients of TCM have problems such as poor water solubility and stability, poor tissue permeability, rapid clearance in the body and short half-life, which leads to limited accumulation in target tissues, greatly restricting clinical application and in-depth research. In the field of tumor treatment research, most of the current focus is on the characteristics of TCM monomers themselves and the role of nanodelivery systems in optimizing their pharmacokinetics (*Lai et al., 2022*). This article puts forward a perspective that has not been adequately explored-the nano TCM monomer delivery system can overcome the shortcomings of TCM monomers in anti-tumor treatment, enabling them to exhibit a stronger anti-tumor effect compared to free monomers. This approach can break through the limitation of focusing solely on a single target or local action mechanism, providing a brand-new and more comprehensive perspective for understanding the mechanism of action of TCM monomers in anti-tumor therapy. It is expected to open up new strategies and approaches for tumor treatment.

## Nanotechnology improves the bioavailability of TCM monomers (*Fu et al., 2021*)

The active ingredients of TCM need to reach a certain blood concentration in the body to play a role, but the absorption and utilization of most poorly soluble ingredients in the preparation of dosage forms after oral administration cannot achieve the expected effect, so the solubility and stability of the drug should be increased as much as possible. Most TCM monomers have low bioavailability and efficacy due to their poor solubility and inability to pass through the cell lipid membrane (*Tian et al., 2021*). The particle size of nano TCM carriers has a great influence on the efficacy of drugs. When the particle size of nanoparticles is smaller, its specific surface area will increase and it will be more soluble in the medium, thereby increasing the dissolution and utilization of drugs in the body (*Alharbi et al., 2021*). In recent years, more and more studies have shown that nanotechnology can effectively improve the bioavailability of TCM monomers. In order to improve the bioavailability of resveratrol, *Wang et al. (2019)* prepared resveratrol PEG-PLGA nanoparticles, and then evaluated their anti-colitis effect *in vitro*. The results showed that nanoparticles enhanced the inhibitory effect of resveratrol on the proliferation of colon cancer cells and improved the bioavailability of resveratrol. *He et al. (2005)* conducted a comparative study on the oral absorption of silymarin solid lipid nanoparticles of different particle sizes in rats and found that the bioavailability of 150 nm solid lipid nanoparticles was significantly higher than that of two preparations with particle sizes of 500 and 1,000 nm. The particle size has a significant effect on the oral absorption of silymarin. The use of nanotechnology to process TCM can break the cell wall of plants and facilitate the exudation of effective ingredients. *Yuan et al. (2013)* prepared cell-penetrating peptide-coated celastrol (CT)-loaded nanostructured lipid

carriers-nanostructured lipid carriers (CT-NLCs). Compared with celastrol free drugs, CT-NLCs can significantly enhance the anti-prostate cancer activity *in vitro* and significantly improve the bioavailability of drugs.

## Nanotechnology improves the targeting of TCM monomers

Targeting refers to the ability of drugs to selectively and purposefully reach specific parts, which can enhance the efficacy of drugs and avoid the side effects caused by systemic diffusion of drugs (*Hristova-Panusheva et al., 2024*). In the early 20th century, Paul Ehrlich proposed the concept of targeted drugs, which consists of three parts: drugs, targeting parts and drug carriers. In the clinical treatment of tumors, targeted drug delivery methods are selected to deliver drugs to specific tumor tissues, cells or organelles to overcome the bottlenecks of aimless distribution of drugs after entering the body, toxic side effects on normal tissues and high-dose administration (*Li et al., 2023*). It can be seen that the drug delivery system loaded with TCM monomers for the treatment of tumors by nanomaterials shows great potential. The targeting of nanomedicines enables the effective ingredients to specifically bind to the lesion site, thereby significantly improving the therapeutic effect of TCM. On the one hand, nano TCM monomers can be designed as a targeted drug delivery system, and on the other hand, the introduction of messenger drugs is conducive to synergistically enhancing targeting.

### *Targeted drug delivery system*

The drug delivery system must meet two conditions: minimal loss in the blood to achieve characteristic site targeting and killing tumor cells without side effects on the body. Once the nanoparticles enter the body, they will be quickly wrapped by the regulatory proteins in the blood to form large aggregates. The regulated nanoparticles can be recognized by the reticuloendothelial system (RES) or the mononuclear phagocyte system (MPS), which is composed of macrophages associated with the spleen and liver (*Dutta, Barick & Hassan, 2021*). The above physicochemical properties of nanocarriers usually determine their fate in biological systems, such as whether they are cleared by the body's clearance system or internalized by tumor cells to trigger biological responses (*Liu et al., 2020*). Internalization includes two methods: passive targeting and active targeting. (1) Passive targeting: Due to the high permeability and poor lymphatic flow of tumor blood vessels, they have an enhanced permeability and retention effect (EPR), which is the main driving force of passive targeting (*Wu, 2021*). This phenomenon is conducive to the targeted entry of nanocarriers into tumor vascularized areas through leaky vascular systems such as tumors, infections, and inflammations *via* oral or intravenous injection. Vascular leakage can increase the concentration of nanocarriers in target tissues, and damaged lymphatic systems can increase the aggregation of nanocarriers in tumor vascularized areas (process as shown in Fig. 1), allowing the nano-drug delivery system to exist in tissues for a long time (*He et al., 2022*; *Wang et al., 2023b*). However, some tumors do not have the EPR effect, and passive targeting still has some limitations. (2) Active targeting: Through the binding of specific ligands to receptors (such as ligand-receptor interaction, antibody-antigen interaction), drugs are selectively delivered to specific target areas (*Yang*

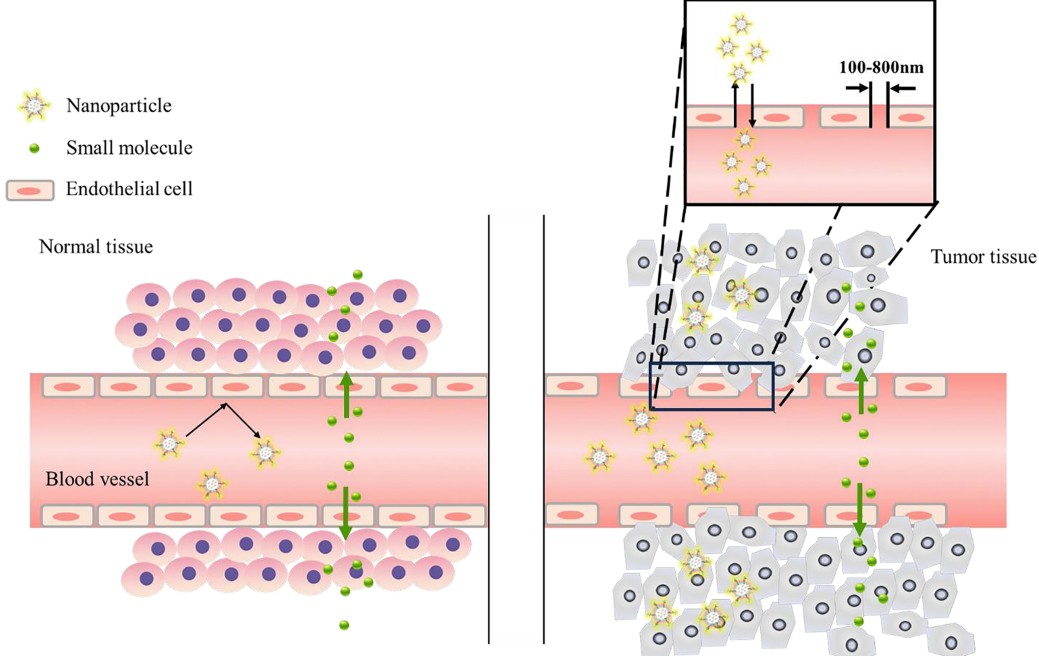

**Figure 1 Enhanced permeability and retention (EPR) effect.** Normal blood vessels are surrounded by a smooth muscle cell layer, and the intercellular junctions are tight, making it difficult for macromolecular drugs to extravasate. In contrast, in tumor tissues, blood vessels always have loose intercellular junctions, and macromolecular drugs can escape into tumor tissues through these junctions. In addition, defects in the lymph node system in tumors cause macromolecular drugs to be retained in tumor tissues.

*et al., 2021*). This method can directly target the corresponding site while avoiding related systemic adverse reactions. It is initiated by the specific interaction between ligands and extracellular receptors. For example, antibodies, nucleic acid chains, peptides and other ligands on the surface of nanomaterials are bound to receptors on the surface of tumor cells, which greatly improves the specificity of recognizing tumor cells. *Hu, Zhou & Zhang (2010)* prepared a novel lactosylnorcantharidin nanoparticle (Lac-NCTD-NPs) encapsulating a new compound, lactosylnorcantharidin (Lac-NCTD). The galactose residues on lactosylnorcantharidin (Lac-NCTD) can be specifically recognized by the asialoglycoprotein receptor (ASGPR) on the surface of hepatocytes *in vivo*, achieving active liver targeting.

### Synergistic enhancement of targeting

In brain targeting, the blood-brain barrier (BBB) makes drug treatment of brain diseases complicated. The blood-brain barrier refers to the barrier between plasma and brain cells formed by the capillary wall and glial cells, and the barrier between the choroid plexus and cerebrospinal fluid, which can help the brain resist foreign substances in the blood (*Boyé et al., 2022*). The blood-brain barrier strictly limits the types of molecules that pass through, but it also prevents drugs and macromolecules from entering the brain. While the BBB absorbs the nutrients needed by brain cells to ensure nutritional balance in the brain, it also excretes metabolites. This material exchange under physiological conditions enables

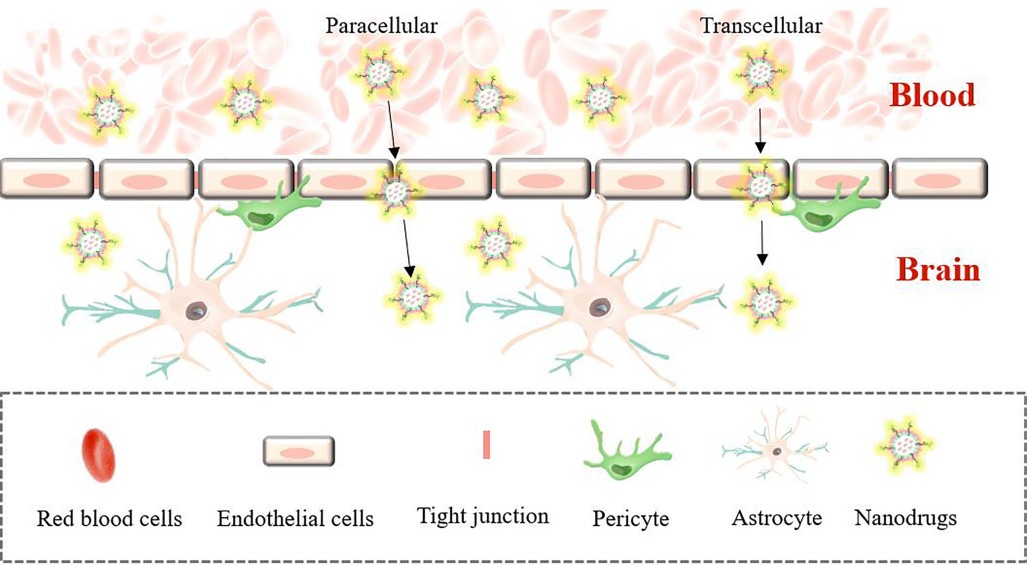

**Figure 2 Brain targeting across the BBB.** Transcellular transport requires energy participation, and the whole process is slow. Paracellular pathways are mostly passive, and transport is achieved mainly by controlling the opening and closing of paracellular pathways through dynamic changes in cell structure.

drug design to have a special transport mechanism, including mainly transcellular membrane channel transport and cell paracellular channel transport (the process is shown in *Fig. 2*). Borneol and musk belong to aromatic TCMs in natural medicines, and their effects are equivalent to targeted preparations. *Qi et al. (2022)* prepared a liposome (Lf-LP-Mu-DTX) loaded with docetaxel (DTX) modified with musk ketone (Mu) that can cross the blood-brain barrier and target the brain. Compared with unmodified liposomes, Lf-LP-Mu-DTX is concentrated in the brain glioma area, and with its own lipid solubility advantage, it enhances the penetration of BBB into the brain and enhances the brain protection effect of the drug.

## Nanotechnology controls the release of TCM monomers

Nanoformulations control the release of TCM monomers mainly in the form of sustained release and responsive triggered release under specific conditions (*Sun & Jiang, 2023*). Sustained release means that the nanosystem loaded with TCM monomers enters the human body and releases the drug evenly, slowly and continuously at a specific targeted site, avoiding premature or sudden release of the drug in large quantities, reducing blood drug concentration fluctuations, and achieving the purpose of sustained release and long-term effect (*Dong et al., 2021*). Based on this principle, a responsive TCM nano-drug delivery system is designed by researchers, which triggers the disintegration, instability, isomerization, polymerization or aggregation of nanocarriers under internal or external stimulation from various physical, chemical or biochemical sources, and releases drug molecules at appropriate concentrations at the target site, thereby leading to controlled release of the drug, avoiding being attacked by the human immune system, and making them more intelligent.

## Nanotechnology inhibits multidrug resistance of tumor

Tumor multidrug resistance (MDR) refers to the phenomenon that tumor cells develop tolerance to one or more chemotherapy drugs, leading to drug treatment failure. The mechanisms of MDR include the heterogeneity of tumor cells, changes in the tumor microenvironment, and overexpression of cell membrane transport proteins. The heterogeneity of tumor cells is the basis for tumor resistance to drugs (*Goyette, Lipsyc-Sharf & Polyak, 2023*). Different subpopulations of tumor cells have different sensitivities to drugs. Some cells can survive under the action of drugs and gradually develop into drug-resistant populations. Changes in the TME also play a key role. For example, hypoxia, acidic environment and changes in extracellular matrix components can affect the response of tumor cells to drugs and promote the formation of drug resistance. *Guo et al. (2024)* prepared a natural polysaccharide-based nanoplatform (TDTD@UA/HA micelles) with dual targeting capabilities for both cells and mitochondria, which was used to co-deliver ursolic acid (UA) and doxorubicin (DOX) for combined treatment of multidrug resistance (MDR). P-glycoprotein (P-gp) is the most typical cell membrane transporter. Drug-resistant tumor cells will produce a large number of membrane transporters such as P-gp on the membrane surface that can expel drugs from the cell, while preventing drug intake, resulting in a significant reduction in the total amount of drugs in the cell. *Zhang et al. (2023)* prepared pH-sensitive liposomes (LPs) encapsulating dihydroartemisinin (DHA) and tetrandrine (TET). Under acidic conditions, the drug release rate significantly increased. By enhancing the cellular uptake capacity, the cytotoxic effect on tumor cells was strengthened. DHA and TET were encapsulated within the nanoparticles. By inhibiting the p53 (R248Q)-ERK1/2-NF-κB signaling pathway, the expression of the CIZ1 gene associated with TGF-β1/Smad signal transduction was down-regulated. Subsequently, the expression of P-gp was inhibited, reducing drug efflux and decreasing the drug resistance of tumor cells.

Nano drugs have shown great potential in overcoming the MDR problem, especially after loading TCM monomers. Enhanced permeability and retention (EPR) and targeting ligands can be used to increase the accumulation of drugs in tumor sites and reduce the toxic side effects of normal tissues. The shielding effect of nanocarriers can also be used to avoid being recognized and excreted by transporters and increase the concentration of drugs in cells. Nanocarriers have endocytosis, which can promote the entry of drugs into cells and release them through endosomal escape or dissolution to avoid lysosomal degradation. The combination of nanotechnology and TCM monomers has opened up a new way to improve the efficacy and safety of TCM monomers in anti-tumor treatment.

## ANTI-TUMOR STRATEGY OF NANO-CARRYING TCM MONOMERS TECHNOLOGY

So far, although there are many types of drug treatments for cancer and there are differences in functional components, most researchers mainly focus on the therapeutic effects of drugs on tumor cells and TME.

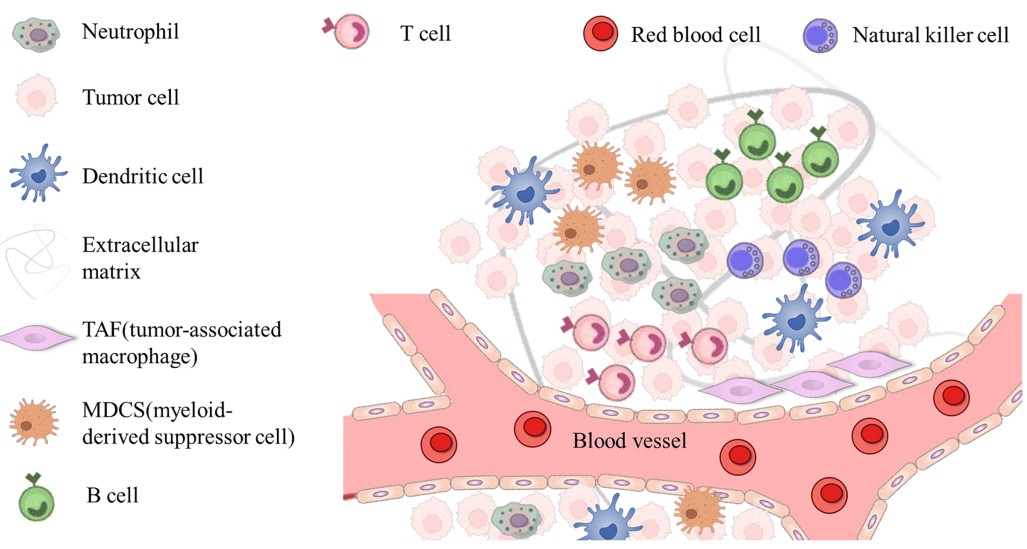

**Figure 3 The main components of the tumor microenvironment.** The cellular environment in which tumor cells reside is called the tumor microenvironment, which includes neutrophil, tumor cell, dendritic cell, TAF (tumor associated macrophage), MDCS (myeloid-derived suppressor cell), red blood cell, immune cells (T and B cells, *etc.*) and extracellular components (extracellular matrix, extracellular matrix, *etc.*), which surround tumor cells and are nourished by the vascular system.

## Inhibition of tumor cell proliferation and metastasis

Tumor cells have the characteristics of unlimited proliferation. TCM monomers can inhibit tumor cell proliferation by promoting tumor cell apoptosis, inducing cell autophagy, enhancing oxidative stress, promoting sensitization and blocking cell cycle, thereby achieving the effect of treating tumors (*Srimanta et al., 2021*).

Tumor cell metastasis is the main feature of malignant tumors and the primary factor causing death in cancer patients. Tumor cells release various proteolytic enzymes to destroy the tissues at their adhesion sites, that is, to destroy the basement membrane of the extracellular matrix and the vascular wall, to achieve infiltration metastasis and the formation of new blood vessels (*Zhao et al., 2023b*; *Ma et al., 2021*).

## Therapeutic strategies of nano TCM monomers in the tumor microenvironment

TME refers to the microenvironment surrounding tumor cells, which is composed of surrounding blood vessels, immune cells, fibroblasts, myeloid inflammatory cells, various signaling molecules and extracellular matrix. The structure is shown in Fig. 3. Its unique physiological and pathological characteristics have a profound impact on the occurrence, development, metastasis and treatment response of tumors (*de Visser & Joyce, 2023*; *Zhao et al., 2023a*). In recent years, drug delivery systems loaded with traditional TCM monomers have played a significant role in interacting and regulating with TME. Therefore, in-depth exploration of the mechanism of their interaction is of great significance for tumor treatment. The size, shape and surface charge of nanomaterials

determine the action pathway in TME. Nanoparticles are more likely to pass through abnormal blood vessel walls in tumor tissues due to their smaller particle size, thereby achieving passive targeted enrichment (*Manikkath et al., 2023*). At the same time, surface modification of nanomaterials can regulate their interaction with various components in TME. For example, by modifying hydrophilic polymers such as polyethylene glycol (PEG), the nonspecific adsorption of nanoparticles in the blood circulation can be reduced, their circulation time in the body can be prolonged, and the chance of reaching the tumor site can be increased (*Naito et al., 2022*). The TME has the characteristics of low pH, high reactive oxygen species (ROS) levels, and abnormal angiogenesis. Therefore, these characteristics can be used to achieve intelligent responsive release using nanomaterial-loaded TCM monomer delivery systems (*Zhu et al., 2023*). In the TME with low pH and high ROS levels, nano-TCM can trigger the responsive nanomaterials to release drugs, allowing the drugs to act more precisely on the tumor site.

From a cellular level, immune cells in the TME play a key role in tumor immune surveillance and escape. Nanomaterial-loaded TCM monomer delivery systems can regulate the functions of immune cells. On the one hand, some TCM monomers can activate immune cells, such as macrophages and T cells. The delivery of nanomaterials can increase the uptake of TCM monomers in immune cells and enhance their immunoregulatory effects (*Park et al., 2023*). On the other hand, there are immunosuppressive cells in the TME, such as regulatory T cells (Tregs) and myeloid-derived suppressor cells (MDSCs). Nanomaterials inhibit the activity of these immunosuppressive cells by delivering TCM monomers with immunomodulatory functions, thereby reversing the immunosuppressive microenvironment and restoring the body's anti-tumor immune response (*Mao et al., 2021*). Tumor-associated fibroblasts (CAFs) are an important component of the TME. They secrete a large amount of extracellular matrix components that affect the migration and invasion of tumor cells. These extracellular matrices provide physical support for tumor cells and interact with cell surface receptors to affect cell signaling (*Siska et al., 2020*). The nanomaterial-loaded TCM monomer delivery system may inhibit the metastasis of tumor cells by regulating the function of CAFs and changing the composition and structure of the extracellular matrix. Nanomaterials protect TCM monomers from passing through the ECM to reach the TME, thereby regulating ECM metabolism and reducing the migration of tumor cells.

Tumor angiogenesis also has an impact on tumor growth and metastasis. The vascular structure and function in the TME are abnormal. The nanomaterial-loaded TCM monomer delivery system can affect the formation of tumor blood vessels by regulating angiogenesis-related signaling pathways. On the one hand, TCM monomers may inhibit the expression or activity of angiogenic factors such as vascular endothelial growth factor (VEGF) and reduce the formation of new blood vessels (*Zou et al., 2020*); on the other hand, nanomaterials can improve the permeability of tumor blood vessels and enhance the delivery efficiency of drugs, so that more TCM monomers can reach tumor cells (*Zeng et al., 2023*).

# MAIN TYPES OF TCM NANO-DRUG DELIVERY SYSTEMS AND THEIR APPLICATIONS IN ANTI-TUMOR

## Lipid nanomaterials

Amphiphilic phospholipids form a bilayer structure, and cholesterol supports and maintains the bilayer structure. According to the pharmacokinetic properties, this structure can contain both hydrophilic and hydrophobic drugs (*Amiri et al., 2023*). Therefore, the TCM monomers wrapped inside the liposomes can increase their solubility and avoid being degraded by the environment during the blood circulation of the human body.

Liposomes improve the stability and biocompatibility of TCM monomers with poor water solubility by loading them. Curcumin is a poorly soluble TCM monomer with a diketone structure in its structure, which can be complexed with metal ions to form a stable poorly soluble complex. *Liu et al. (2021)* invented a method of using metal ions as the inner water phase to prepare liposomes with a transmembrane metal ion gradient to actively load curcumin, thereby achieving effective drug delivery. Quercetin is a TCM monomer which has a significant effect of inhibiting EMT, thereby inhibiting tumor growth. Quercetin can induce apoptosis of Y79 cells through the JNK and P38MAPK pathways, and inhibit the multidrug resistance of tumor cells by downregulating p-gp expression. However, the poor solubility of quercetin limits its use. *Lin et al. (2024)* loaded the poorly soluble quercetin and the hydrophilic doxorubicin into the lipid shell and the aqueous core of the liposome to form QD Lipo. The extremely high compatibility of the liposome improved the internalization of quercetin into Y79 cells. When mitochondria are overproduced, EMT is activated. QD Lipo reduces the expression level of ROS and significantly downregulates Vimentin and α-SMA proteins, indicating that the prepared AD Lipo can enhance the tumor-killing effect of quercetin. Liposomes can also improve the pharmacokinetics and tissue distribution of TCM monomers, reduce the toxicity of TCM monomers, and improve the therapeutic index. *Xia et al. (2022)* prepared Rg3-Lp/DTX by thin film hydration method, which inhibited the NF-κB signaling pathway, reduced the expression of anti-apoptotic protein Bcl2, increased the expression of pro-apoptotic protein Bax, and enhanced the toxicity and apoptotic effect of DTX on tumor cells. At the same time, Rg3 inhibited STAT3 activation, reduced tumor cell secretion of CCL2, reduced the recruitment of MDSCs and TAMs in the lungs, destroyed the metastatic microenvironment, inhibited tumor cell growth and metastasis, increased tumor cell apoptosis, and reduced tumor size.

However, liposomes have the disadvantages of being easily degraded and easily cleared by macrophages in the body, resulting in a short maintenance time in the body. The drugs encapsulated by biomodified liposomes have longer circulation time, stronger targeting and less toxicity. Its mechanism of action is mainly to avoid rapid phagocytosis of the reticuloendothelial system (*Almeida et al., 2020*). Erythrocytes are highly biocompatible, rich in content, and have long circulation. Since they come from the body, they can avoid being cleared by macrophages and increase the uptake of their modified nanomaterials by tumor cells. *Zhong et al. (2021)* prepared a triptolide-erythrin co-loaded liposome, and

then wrapped the erythrocyte membrane outside to make this biomembrane biomimetic drug delivery system to promote the delivery of triptolide in tumors. Liposomes accumulate in tumor sites through the EPR effect with the help of nano-scale particle size; their unique "core-shell" structure has sustained-release properties and can maintain drug efficacy for a long time; compared with free triptolide, liposomes are more easily taken up by tumor cells, and the inhibitory effect of C+TRB CMlip on tumor cells is stronger than the combination of C+T/Lip and free drugs, and can more effectively inhibit the growth and proliferation of tumor cells.

## Exosomes

Extracellular vesicles (EVs) in bilayer phospholipids are nanoscale membrane vesicles secreted by cells, and their size is generally between 50–1,000 nm. EVs are recognized as a natural drug delivery system with low immunogenicity, strong homologous targeting ability, and high biocompatibility. They can be used as a good drug delivery carrier in tumor treatment. According to their origin, EVs can be divided into three categories: exosomes, microvesicles and apoptotic bodies (Wang et al., 2024b). Exosomes are nanoparticles of 40–200 nm, and their functions depend on the cells of origin (Fig. 4). Since exosomes contain lipids and molecules similar to those of their origin cells, exosome nanoparticles can escape immune surveillance and be smoothly internalized with target cells. Exosomes contain a variety of bioactive substances such as proteins, DNA, microRNA, long non-coding RNA, messenger RNA, tumor genes and transcription factors (Negri et al., 2020). The levels of these bioactive substances are related to tumor invasion and TME.

As endogenous substances, exosomes not only have cancer-promoting properties, but also can be used to make them an effective way of anti-tumor treatment by using their immunomodulatory and tumor microcirculation remodeling properties.

TCM can inhibit the development of tumor cells by affecting the intercellular communication of exosomes and the immunomodulatory effect of the tumor microenvironment. Tumors can use the spatiotemporal characteristics of tumor-associated macrophages (TAMs) migration and differentiation in tumor tissues to dynamically educate them, polarizing them into a series of TAMs with different degrees of activation (Liu et al., 2021). Tumor-derived exosomes can regulate the tumor microenvironment by transferring miRNA to immune cells. Epigallocatechin gallate (EGCG) has a strong anti-tumor effect both *in vitro* and *in vivo*. Nie et al. (2019) studied the anti-tumor mechanism of resveratrol against glioblastoma multiforme (GBM) using two GBM cell lines. The anti-tumor mechanism of resveratrol on GBM was studied, and it was found that exosomes secreted by U251 cells (U251/N/Exo) were rich in the pro-apoptotic protein H2AX, which could significantly enhance the sensitivity of LN428 cells to resveratrol. However, the level of H2AX in U251 exosomes (U251/Res/Exo) treated with resveratrol decreased, while the levels of anti-apoptotic proteins Rap1 and F-actin increased, resulting in the loss of its ability to reverse LN428 resistance. This suggests that the protein composition in exosomes can affect the sensitivity of GBM cells to resveratrol by regulating cell apoptosis-related signaling pathways. Zhang et al. (2022) found that

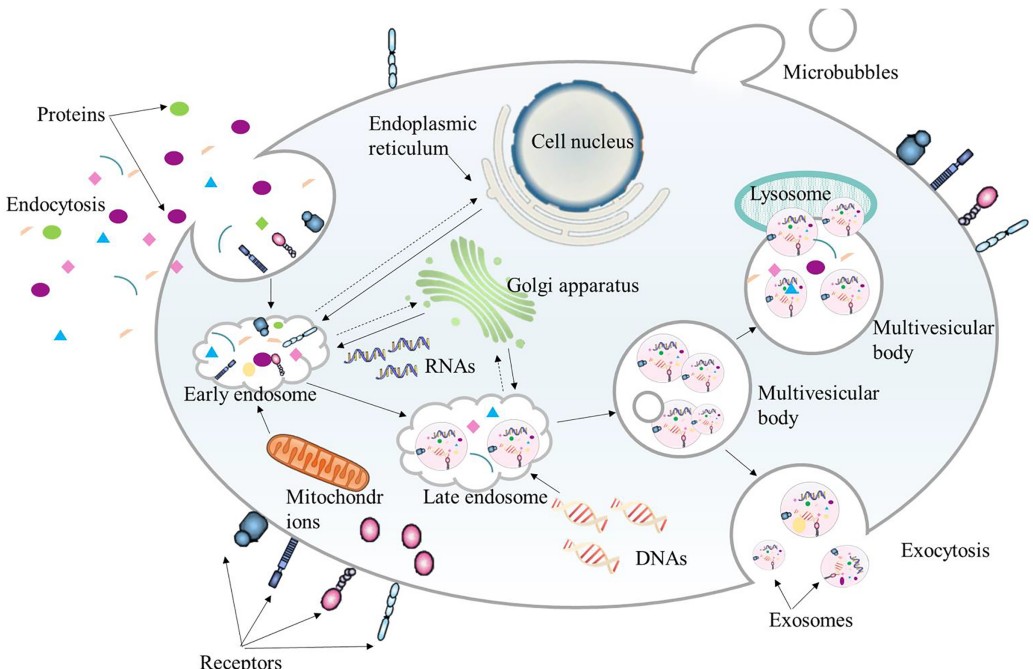

**Figure 4 The biogenesis process of exosome.** Exosomes originate from the endocytosis of cells. They first bud from the plasma membrane to form early endosomes. Then the early endosomes sprout intraluminal vesicles to develop into late endosomes, which further form multivesicular bodies. Finally, multivesicular bodies fuse with autophagosomes and lysosomes to be degraded or fuse with the plasma membrane. The intraluminal vesicles released by the multivesicular bodies fused with the plasma membrane are the final exosomes.

curcumin can be delivered to recipient liver cancer cells through exosomes in tumor treatment. Experiments have shown that curcumin exosomes can enter HepG2 cells through endocytosis and inhibit their vitality. They regulate the HepG2 cell cycle, significantly increase the proportion of cells in the S phase and G2/M phase, significantly decrease the proportion of cells in the G0/G1 phase, block the cell cycle in the G2/M and S phases, and inhibit cell proliferation. At the same time, curcumin exosomes induce apoptosis of HepG2 cells. It can upregulate the expression of the pro-apoptotic protein BAX and downregulate the expression of the anti-apoptotic protein Bcl-2. Through the mitochondrial apoptosis pathway, it breaks the mitochondrial membrane homeostasis, promotes the release of cytochrome C, activates the caspase cascade reaction, and finally exerts an anti-tumor effect. In addition, cell cycle arrest may increase the sensitivity of cells to apoptotic signals. The two synergistically inhibit tumor cell growth.

TCM can also inhibit tumor formation and metastasis by inhibiting the secretion of tumor-derived exosomes (TDEs). *Wang et al. (2020a)* used RT-qPCR to detect that exosome delivery promoted verbbascoside to deliver miR-7-5p to recipient glioblastoma (GBM) cells in subcutaneous tumors and metastatic tumors in mice.The results showed that verbascoside can promote the expression of miR-7-5p in glioblastoma (GBM) and promote its delivery to recipient GBM cells. MiR-7-5p inhibits GBM cell proliferation, migration, invasion and microtubule formation by inhibiting the epidermal growth factor

receptor (EGFR)/phosphatidylinositol 3-kinase (PI3K) protein kinase B (PKB/Akt) signaling pathway, and can inhibit GBM tumorigenesis and metastasis *in vivo*, thereby inhibiting the occurrence of GBM (Fig. 5). *Tong et al. (2024)* explored the effect of resveratrol on exosome secretion and the role of resveratrol-induced exosomes in the progression of hepatocellular carcinoma. It was found that resveratrol inhibited exosome secretion by downregulating the expression of Rab27a, thereby inhibiting the proliferation, migration and epithelial-mesenchymal transition of Huh7 cells. In addition, resveratrol-induced exosomes can also inhibit the malignant phenotype of Huh7 cells by inhibiting the nuclear translocation of β-catenin and the activation of autophagy mediated by lncRNA SNHG29. *Du et al. (2020)* found that the total steroidal alkaloid glycosides in *Solanum lyratum* Thunb. The experimental results show that exosomes secreted by A549 cells can promote the transformation of HUVECs into tumor-associated endothelial cells (Td-ECs), while steroidal glycoalkaloids (such as TSGS and SA1) can significantly inhibit this transformation. This indicates that Steroidal glycoalkaloids from *Solanum lyratum* can directly affect the formation of tumor exosomes by agglutinating tumor cell membrane lipid cholesterol, leading to changes in their function, thereby inhibiting tumor angiogenesis and inhibiting the development of lung cancer A549 cells.

There are many studies on the role and related mechanisms of tumor-derived exosomes in the occurrence, development, invasion and metastasis of lung cancer, but There are still many problems to be solved by researchers in the study of exosomes, such as the purification and labeling methods of exosomes, how to find the target genes of exosomes, the mechanism of action and signal pathways of exosomes, *etc*. In short, the study of exosomes has broad prospects.

## Polymer micelles

The particle size of polymer micelles is in the range of 10–100 nanometers. They are not easily recognized and captured by the endothelial reticular system res in the blood circulation. The nano drug delivery system can exist stably and for a long time in the blood, and at the same time achieve passive targeting to the tumor site through the EPR effect (*Zheng et al., 2020*). The solubility of conventional polymers will decrease after loading hydrophobic drugs, while polymer micelles still have good water solubility after loading a large amount of hydrophobic drugs. This is because the distribution coefficient of hydrophobic drugs in the hydrophobic core of polymer micelles is high, which can achieve a higher drug loading and encapsulation rate, while hydrophilic drugs are encapsulated. The shell-core interface can load amphiphilic drugs and widely deliver anti-cancer drugs (*Thotakura, Parashar & Raza, 2021*).

Polymer micelles can promote intracellular drug accumulation and anti-tumor effects. SMVT, the main transporter of biotin, is overexpressed in many cancer cells (such as lung cancer cells) and mediates the internalization of biotin-modified micelles. *Wang et al. (2020c)* prepared a poly(HPMAm)-based biotin-modified micelle loaded with paclitaxel for anti-tumor nano-drug delivery system. Biotin-modified micelles bind to biotin receptors (SMVT) on the surface of tumor cells, enter cells through receptor-mediated endocytosis, release paclitaxel (PTX) to exert cytotoxicity, and non-cancerous cells

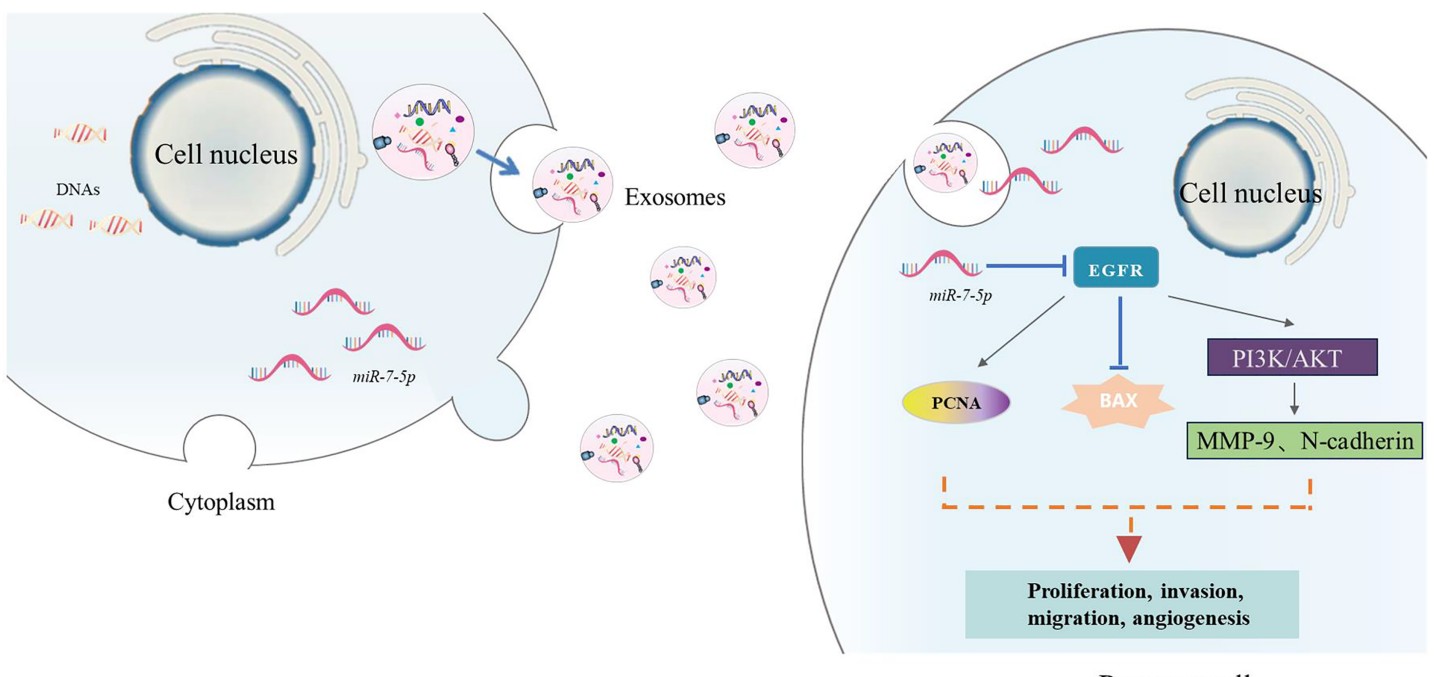

**Figure 5 The action process.** EGFR (epidermal growth factor receptor), PCNA (proliferating cell nuclear antigen), Bax (B-cell lymphoma-2 associated X protein), MMP-9 (matrix metalloproteinase-9 N-cadherin-neurocadherin), N-cadherin (Neural-cadherin/Cadherin-2), N-cadherin (Neural-cadherin/Cadherin-2), PKB/Akt (protein kinase B), PI3K (phosphatidylinositol-3-kinase).

(HEK293 cells, biotin receptor negative) have low uptake of micelles. *Li et al. (2021)* prepared polymer micelles (SK/ siIDO1-HMS) loaded with shikonin (SK) and IDO-1 knockout siRNA (siIDO1). The results showed that SK/Siido1-HMS could trigger immunogenic cell death (ICD), promote CRT exposure, increase the DC maturation rate in the draining lymph nodes to 46.9 ± 7.3%, increase the infiltration of CD8⁺ T cells in the tumor, increase the concentration of γ-interferon (Interferon-γ, IFN-γ) and tumor necrosis factor-α (Tumor Necrosis Factor-α, TNF-α) in the tumor, reduce the expression level of IDO-1 and the ratio of kynurenine to tryptophan, reduce the frequency of infiltrating regulatory T cells (Treg) in the tumor, and increase the anti-tumor effect compared with free monomers.

Since the optimal drug concentration is required at the tumor site and in the cell to effectively kill the cell, and the hydrophilic shell of the micelle can be modified, it is necessary to effectively release the drug to the tumor site by introducing a stimulus-responsive prodrug. At present, the most widely studied is pH-responsive micelles. *Liang et al. (2023)* designed an amphiphilic polymer micelle (p(mPEG-co-HPBE-co-EMD)CLB) with dual reactive oxygen species (ROS)/pH responsiveness to enhance the anti-tumor effect. In the polymer experiment, the proliferation of 4T1 breast cancer cells was significantly inhibited, and the tumor volume was significantly reduced in the *in vivo* experiment. By consuming intracellular glutathione (GSH) and increasing the level of ROS, the antioxidant system of tumor cells is destroyed, the oxidative stress response in the cells is activated, and then apoptosis is induced. In addition, the structure of the polymer is

destroyed in the tumor microenvironment, releasing the drug and enhancing the cellular uptake and accumulation of the drug, further improving the anti-tumor effect. *Sun et al. (2020)* designed an enzyme-sensitive core-targeted bifunctional polymer micelle (9-NC/HATPC) to enhance the anti-tumor effect of 9-nitrocamptothecin (9-NC). The micelle achieves enzyme responsiveness through the cathepsin B-sensitive ALAL peptide segment, and forms a positively charged secondary micelle (9-NC/TPC) after cleavage, which promotes lysosomal escape and enters the cell nucleus with the help of TAT peptide, inhibits topoisomerase I, and induces tumor cell apoptosis. The micelle increases the drug loading capacity through π-π stacking to ensure stable drug delivery, while using the acidic environment and enzyme activity in tumor cells to achieve targeted release, enhancing the cellular uptake and nuclear accumulation of drugs. In recent years, although many polymer micelle drug-loaded samples have entered the clinical trial stage, the toxicological and pharmacological properties of the anti-tumor active ingredients of traditional Chinese traditional medicine are not yet fully understood, and the delivery of anticancer drugs by polymer micelles still faces many challenges, such as its potential toxicity, the potential mechanism between the stability of micelles and drug delivery is not yet fully understood, and how micelles interact with drugs.

## Carbon nanotubes

Carbon nanotubes have extremely high tensile strength and elastic modulus. Their structure is a sp hybridized hexagonal arrangement. They are one of the hardest materials in the world and can conduct heat and electricity. Carbon nanotubes are generally divided into single carbon nanotubes (SWCNTs) and multi-carbon nanotubes (MWCNTs). Single carbon nanotubes consist of a single cylindrical graphene, covered with two segments in a hemispherical arrangement of a carbon grid (*Qian et al., 2021*). Multi-carbon nanotubes consist of several to dozens of concentric cylindrical graphene shells with higher lumen filling (*Mohd Nurazzi et al., 2021*). Due to its hollow physical structure and chemical properties that are easy to modify on the surface, carbon nanotubes can be developed based on these excellent characteristics for the delivery of therapeutic drug carriers to enhance the pharmacological activity of TCM monomers, while reducing the side effects of drugs on the body, and increasing the accumulation of drugs at the tumor site.

The inherent stability and structural flexibility of single carbon nanotubes (SWNTs) can prolong the blood circulation time of drugs and improve the bioavailability of drugs. *Mohammadi et al. (2024)* studied the application of platinum-modified carbon nanotube (CNT-Pt-CUR) nanosystem loaded with curcumin (CUR) in tumor treatment. CUR has antioxidant and anti-inflammatory properties, but its poor water solubility limits its application. By loading CUR onto CNT-Pt, efficient drug delivery and tumor microenvironment-responsive release were achieved. Experiments showed that the drug release rate of CNT-Pt-CUR in an acidic environment (pH 4.7) was significantly higher than that in a physiological environment (pH 7.4), showing good pH sensitivity. In addition, combined with X-ray radiation, CNT-Pt-CUR significantly enhanced the killing effect on tumor cells. In PC-3 cells, SWCNT-Cur showed a higher inhibitory effect compared with natural curcumin. In view of the shortcomings of plumbagin in anti-tumor

applications, *Zhong (2011)* covalently coupled water-soluble SWNTs modified with PEG to plumbagin to prepare a stable complex (PLB-PEG-SWENTs). The experimental results showed that after PLB-PEG-SWENTs were internalized into cells, the amide bonds and ester bonds connecting the drug molecules and SWNTs were hydrolyzed and broken, directly releasing the drug molecules to act directly on the G2/M phase control point of tumor cells to initiate the cell apoptosis pathway, thereby affecting cell division and proliferation. Ginsenoside Rg3 is a component of puffed ginseng with anticancer activity. *Luo, Wang & Ji (2021)* prepared multi-walled carbon nanotubes (Rg3-MWCNT) loaded with ginsenoside Rg3. The study found that Rg3 promoted apoptosis of triple-negative breast cancer (TNBC) cells. Rg3-CNT treatment enhanced the phenotype within T cells, reduced interferon γ-induced PD-L1 upregulation in breast cancer cells (IFN-γ), and reduced PD-1 expression in activated T cells. This indicates that Rg3-CNT enhances the anticancer effect of Rg3 on TNBC by inhibiting the PD-1/PD-L1 axis.

## Dendritic macromolecules

Dendritic macromolecules form highly bifurcated, monodisperse macromolecules similar to dendritic structures through continuous repetition, growth, and polarization. With the increase of polymerization generations, the degree of polarization continues to expand, and finally a three-dimensional spherical structure consisting of a central core connected to branches, branching units, and external capping groups is formed (*Singh & Kesharwani, 2021*). These special highly branched structures provide functional advantages for the application of dendritic macromolecules. Dendritic macromolecules can prevent drugs from being rapidly enzymatically hydrolyzed, hydrolyzed, or cleared by the immune system, thereby reducing the frequency of administration. Its structural characteristics limit the circulation of drugs in the body and reduce the adverse reactions of drugs, so the dosage can be increased. Its surface is densely covered with various functional groups, and specific functional groups can recognize target cells to achieve targeted drug delivery (*Sharma et al., 2017*). Connecting poorly soluble drugs to the branched structure can greatly improve the solubility of the drug and increase its bioavailability. In short, dendritic macromolecules have the advantages of being well absorbed by cells, clear molecular weight, adjustable branches, multifunctionality, and cavity encapsulation of drugs to improve solubility, making them ideal candidates for delivering anti-tumor drugs.

Different dendrimers have been used in the development of nano-drug delivery systems, such as polyamide, polyamide silicone, polypropylene imine and sugar dendrimers. Dendrimers load drugs through simple encapsulation, electrostatic interaction and covalent coupling. However, so far, only a few active ingredients of traditional TCM, such as puerarin, curcumin, resveratrol, genistein and podophyllotoxin, have been encapsulated in dendrimers. *Khalil et al. (2021)* designed and synthesized a third-generation triazine-based dendrimer and combined it with magnolol and targeting ligands lactobionic acid and folic acid to inhibit matrix metalloproteinases (MMP-2/9) to inhibit the progression of hepatocellular carcinoma (HCC). The results showed that dendrimers and their derivatives inhibited tumor growth by inhibiting MMP-2/9 activity, reducing tumor cell proliferation, invasion and migration. However, the article did not discuss specific signaling pathway

proteins in detail. *Gallien et al. (2021)* verified the antitumor effect of G4 90/10-Cys dendrimers on multiple glioblastoma cell lines. Experiments have shown that G4 90/10-Cys dendrimers can effectively encapsulate curcumin and significantly reduce the activity of human (U87), mouse (GL261) and rat (F98) glioblastoma cells, while having little effect on normal cells such as HEK 293, mouse and rat bone marrow mesenchymal stem cells. Encapsulated curcumin significantly improves the anti-tumor effect by increasing its bioavailability.

PAMAM polyaminoamine dendrimer is one of the most widely studied dendrimers because it is water-soluble and can promote metastasis through the paracellular pathway through epithelial tissue. *Sharma et al. (2019)* prepared a PAMAM-encapsulated antitumor drug podophyllotoxin nano-drug delivery system to evaluate the therapeutic effect of the polymer on hepatocellular carcinoma HCC. In cell experiments, different doses of Opodo significantly reduced the levels of inflammatory factors IL-6 (interleukin-6) and NF-κB (nuclear factor κB). After induction by Dena, liver fibroblast markers TGF-β (transforming growth factor β) and α-SMA (α-smooth muscle actin) were expressed in hepatocellular carcinoma. These proteins play a key role in the progression of HCC, indicating that the development of HCC is through regulating the inflammatory and fibrotic factors DPODO by inhibiting the expression of these proteins. The results showed that the Dpodo group was significantly inhibited after treatment, reducing inflammatory response and liver fibrosis, thereby inhibiting tumor development.

### Other nanoparticles

The delivery system for the encapsulation of anti-tumor active components of traditional TCM also includes nanocages, metal-organic frameworks(MOF), nanohydrogels, and carrier-free nanoformulations. So far, there has been little research on these nanodelivery systems, but their application potential is worthy of attention. *Wu et al. (2022)* studied the therapeutic mechanism of a new type of nanogel (Rhein-DOX nanogel) for liver cancer. The experiment used HepG2 liver cancer cell line and nude mouse subcutaneous transplant tumor model. The results showed that Rhein-DOX nanogel significantly increased the level of intracellular ROS by targeting mitochondria, reduced mitochondrial membrane potential ($\Delta\Psi$m), and induced cell apoptosis. The experimental results show that the nanogel exerts its anti-tumor effect by enhancing ROS-mediated mitochondrial damage and inducing cell apoptosis. Among them, iron-based MOF combines the advantages of MOF and the role of iron in tumor treatment, and has good application prospects in the development of iron-involved multimodal tumor treatment strategies. In particular, it is used to develop iron-involved multimodal cancer treatment strategies. *Wang et al. (2023a)* constructed a BSA-FA functionalized iron-containing metal organic framework (TPL@TFBF). The experimental results showed that TPL inhibited Nrf2 expression and interfered with the *de novo* synthesis of glutathione (GSH), increasing the sensitivity of cells to ferroptosis. At the same time, $Fe^{3+}$ increased the intracellular ROS level through the Fenton reaction after entering the cells. The two synergistically amplified ROS generation and induced ferroptosis, which was manifested as downregulation of Nrf2 and GPX4 expression and decreased GSH levels. On the other hand, TPL can induce

GSDME-dependent cell pyroptosis. $Fe^{3+}$ enhances this process by activating caspase-3, causing it to cut GSDME to produce N fragments, forming pores in the cell membrane, causing cell swelling, membrane rupture, and release of IL-1β and LDH, *etc.*, triggering cell pyroptosis. *Chen et al. (2025)* prepared anisamide-modified ursolic acid nanoparticles, and the experimental results showed that it could significantly downregulate the expression of NRG1 in CAF cells, thereby reducing the phosphorylation levels of HER3 and AKT in LNCaP cells, thereby alleviating enzalutamide resistance. This mechanism suggests that the nanoparticles enhance the sensitivity of prostate cancer cells to enzalutamide by targeting CAF cells and regulating signaling pathways in the tumor microenvironment.

Table 2 summarizes and lists the types of nanomaterials loaded with TCM monomers, and compares the efficacy of different nanomaterials when loaded with TCM monomers and their anti-tumor mechanisms. In terms of detection, dynamic light scattering (DLS) is often used to detect the particle size of nanomaterials, electrophoretic light scattering is used to detect Zeta potential, high performance liquid chromatography (HPLC) or ultraviolet-visible spectrophotometry (UV-Vis) is used to detect drug loading, and dialysis or ultracentrifugation is used to detect encapsulation efficiency. In addition, the characterization of nanoparticles after drug loading is also listed, so as to comprehensively study and analyze the relevant characteristics of nanomaterials loaded with TCM monomers.

## Clinical trial of nano TCM monomer delivery system

In the field of research on nano TCM monomers for anti-tumor effects, only the nano drug delivery system loaded with paclitaxel has been clinically studied. Genexol® PM is the first polymer micelle approved for the treatment of human diseases. It was developed by Samyang Biopharm, a Korean biopharmaceutical company, and is a polymer micelle that encapsulates paclitaxel without polyoxyethylene castor oil (*Gupta et al., 2021*). The polymer material used in this product is methoxypolyethylene glycol-b-poly D, L-lactide (mPEG-b-PDLLA), with a particle size of 20~50 nm and a drug loading of 16.7%. It was first launched in South Korea for the treatment of metastatic breast cancer (MBC), non-small cell lung cancer (NSCLC) and ovarian cancer. In Phase II clinical studies, this product was used alone to treat metastatic breast cancer, or in combination with cisplatin to treat non-small cell lung cancer. Compared with paclitaxel injection and paclitaxel albumin nanoparticles, the efficacy was significantly improved. As a result, Genexol PM was approved in South Korea in 2007 for the treatment of non-small cell lung cancer, metastatic breast cancer and ovarian cancer, and was subsequently launched in India, Serbia, the Philippines and Vietnam. Lipusol (liposomal paclitaxel for injection) was granted by the China National Drug Administration for the treatment of ovarian cancer and breast cancer receiving chemotherapy containing doxorubicin (*Wang et al., 2022*). Lipusol combined with cisplatin can be used to serve patients with non-small cell lung cancer who cannot undergo surgical resection or radiotherapy. Compared with free treatment, lipusol can alleviate the side effects involving blood pressure, peripheral blood, medulla and liver after injection, thereby improving the efficacy of the drug. Albumin-bound paclitaxel (hereinafter referred to as albumin paclitaxel) prepared using

**Table 2** Characterization, efficacy and anti-tumor mechanism of nano-TCM monomers.

| Nanomaterial | Monomer | Experimental model | Changes in tumor cells and tumor size | Particle size | Zeta potential | Stability | Drug loading ratio (DL%) = or drug loading capacity (LC%) | Encapsulation efficiency | Efficacy | Antitumor Mechanism |
|---|---|---|---|---|---|---|---|---|---|---|
| Lip, osomes | Curcumin | Tumor-bearing mice | Inhibits tumor cell activity and subcutaneous tumor growth | 50–100 nm | −25 mV or more | 4 °C for 6 months | LC% = Curcumin: total phospholipids = 0.5–1:10 | 95% or more | Liposomes can encapsulate hydrophilic and hydrophobic drugs, improve the stability, biocompatibility and solubility of traditional Chinese medicine monomers, improve pharmacokinetics and tissue distribution, reduce toxicity, and improve therapeutic index. | Significantly inhibits the growth of subcutaneous tumors in mice and alleviates the progression of metastatic tumors (Park et al., 2023) |
| | Quercetin | Y79 cell | Reduces tumor volume | 108.87 nm | −34.83 mV | 4 °C for 5 days | 1.26% | 96.20% | | Increase the internalization of drugs in Y79 cells and increase the killing effect of drugs on tumors (Mao et al., 2021) |
| | Ginsenoside Rg3 | HepG2 and A549 cell | Reduces the rate at which tumors grow in size | 133.9 nm | −23 mV | | | 82.47% | | Increased the expression of pro-apoptotic protein Bax, and enhanced the toxicity and apoptotic effect of DTX on tumor cells. At the same time, Rg3 inhibited STAT3 activation, reduced tumor cell secretion of CCL2, reduced the recruitment of MDSCs and TAMs in the lungs, destroyed the metastatic microenvironment (Siska et al., 2020) |
| | Triptolide | HepG2 cell | Enhance uptake of HepG2 cells | 119.12 ± 2.78 nm | −16.9 ± 1.2 mV | 4 °C for 7 days | LC% = 0.06 ± 0.002% | 88.1 ± 1.8% | | Celastrol exerts its antitumor effects through mechanisms such as inhibiting cell proliferation, inducing autophagy, and suppressing inflammatory stress. The combined use of the two can exert a synergistic effect by blocking the cell cycle and inducing apoptosis, reducing the drug dose and adverse reactions (Zeng et al., 2023) |
| | Curcumin | A549 cell | Significantly inhibited the invasion of A549 cells, with a tumor suppression rate of 90.3% | 93 nm | +3 mV | | | 95.23% | | It has active tumor targeting, which can significantly improve tumor penetration and cellular internalization in the liposome drug delivery system (Gallien et al., 2021) |
| | Honokiol | A549 cell | Inhibited tumor cell proliferation, tumor inhibition rate of 60.7% | 115.7 nm | −18.31 mV | Room temperature for a week | | | | It enters the cell through endocytosis and can escape from lysosomes to improve drug absorption; CLT induces apoptosis, and apoptosis may be manifested by increased intracellular $Ca^{2+}$ levels through the mitochondrial pathway. CLT can also significantly inhibit the migration and invasion of A549 cells, significantly reduce tumor cell growth, and increase tumor cell proliferation (Ki67 level decreases) and apoptosis (apoptosis rate increased by TUNEL detection) (Sharma et al., 2019) |

(Continued)

| Nanomaterial Monomer | Experimental model | Changes in tumor cells and tumor size | Particle size | Zeta potential | Stability | Drug loading ratio (DL%) = or drug loading capacity (LC%) | Encapsulation efficiency | Efficacy | Antitumor Mechanism |
|---|---|---|---|---|---|---|---|---|---|
| Ginsenoside Rg3 | MCF-7cell Mouse breast cancer model | Reduced the production of tumor-associated fibroblasts (TAFs) and collagen fibers, resulting in a tumor inhibition rate of 90.3% | 88.7 ± 0.99 nm | | 37 °C for 72 h | DL% = 20.15% | 97.35% | | Rg3-PTX-LPs remodel the tumor microenvironment (TME) by inhibiting the IL-6/STAT3/p-STAT3 signaling pathway, repolarizing pro-tumor M2 macrophages to anti-tumor M1, inhibiting the activity of myeloid-derived suppressor cells (MDSCs), and reducing the production of tumor-associated fibroblasts (TAFs) and collagen fibers (Wu et al., 2022) |
| Salvianolic acid B | Breast cancer NIH3T3 cell | Reduces the migration and invasion ability of tumor cells, inhibits tumor growth, and reduces tumor weight | 166.2 nm | −37.76 mV | 4 °C for 15 days | LC% = 17% | 82% | | Enhance penetration in tumors and reshape tumor microenvironment (Chen et al., 2025) |
| Paclitaxel | C6 glioma cell line, C6luc glioma cell line, human cerebral vascular endothelial cell line, HBMEC, orthotopic glioma model | Inhibited the proliferation and migration of C6 cells, induced their apoptosis, significantly inhibited the growth of glioma in situ, and the tumor volume was small. | 128.15 ± 1.63 nm | −18.31 mV | | DL% = 1.52 ± 0.12% | 98.76 ± 0.08% | | RVG15 can be modified to bind to nAChR, enhance the uptake of HBMECs and tumor cells, and mediate endocytosis into cells through clathrin. It has a strong ability to penetrate the blood-brain barrier, and can accumulate within the tumor and penetrate deep into the parenchyma; The released PTX exerts a chemotherapeutic effect, inhibits tumor growth and metastasis, and prolongs the survival time of mice (Wu et al., 2022). |
| Gambogic acid | 4T1 breast cancer cells and human breast cancer cells MDA-MB-231, mouse 4T1 tumor-bearing model | AZGL has inhibitory effect on 4T1 and MDA-MB-231 cells, and the tumor almost disappears. | 87.7 ± 9.2nm | −16.9 mV | 7 days | DL% = 1.04 ± 0.01% | 38.40 ± 0.57% | | GA in AZGL nanoparticles can inhibit HSP90 overexpressed under heat stress, re-sensitization photothermal therapy (PTT); TCPP-mediated photodynamic therapy (PDT) produces ROS to kill heat-tolerant tumor cells; The heat generated by PTT can not only kill cancer cells, but also alleviate tumor hypoxia and enhance the PDT effect. In AZGL, tumors can accumulate, and tumors in the AZGL (PDT PTT) group almost disappear, showing good anti-tumor effects (Wang et al., 2023a). |

# Table 2 (continued)

| Nanomaterial | Monomer | Experimental model | Changes in tumor cells and tumor size | Particle size | Zeta potential | Stability | Drug loading ratio (DL %) = or drug loading capacity (LC%) | Encapsulation efficiency | Efficacy | Antitumor Mechanism |
|---|---|---|---|---|---|---|---|---|---|---|
| Exosomes | Resveratrol | U251 and LN428 GBM cell lines | Promotes apoptosis | 30–200 nm | | | | | It has low immunogenicity, strong homologous targeting, and high biocompatibility to evade immune surveillance. Traditional Chinese medicine exerts anti-tumor activity by regulating TAMs in the tumor microenvironment, regulating tumor cell cycle, inducing apoptosis, inhibiting tumor cell proliferation, migration, invasion and microtubule formation through exosomes, and can also inhibit tumor formation and metastasis by inhibiting the secretion of tumor-derived exosomes | U251 cells are sensitive to resveratrol and exhibit growth arrest and apoptosis, while LN428 cells exhibit drug resistance. It was found that the exosomes (U251/N/Exo) secreted by U251 cells could significantly enhance the sensitivity of LN428 cells to resveratrol, while the U251 exosomes (U251/Res/Exo) treated with resveratrol lost this ability. Protein composition in exosomes may play a key role in regulating the sensitivity of GBM cells to resveratrol (Almeida et al., 2020). |
| | Curcumin | HepG2 cell | | 30–150 nm | | | | | | By inactivating the Wnt/β-catenin pathway and autophagy. In addition, lncRNA SNHG29 plays a key mediating role, silencing activates autophagy, increases β-catenin nuclear translocation to inhibit tumor cell malignant phenotype and mouse tumor growth (Zhong et al., 2021). |
| | Verbascoside | U87 and U251, subcutaneous xenograft model and lung metastasis model | It inhibits the proliferation, migration and invasion of GBM cells, and promotes apoptosis and inhibits tumor growth and metastasis | 64.38 nm | | | | | | VB-treated GBM cells showed significant inhibition of proliferation, increased apoptosis, and decreased migration and invasion. The expression of miR-7-5p in exosomes was up-regulated, and it could be transmitted to recipient cells through exosomes, inhibiting the EGFR/PI3K/Akt signaling pathway. In vivo: In a subcutaneous xenograft model, VB treatment significantly inhibited tumor volume and weight; In the lung metastasis model, VB reduced the number of lung metastases. The expression of miR-7-5p was up-regulated in tumor tissues, while the expression of EGFR, PI3K, p-Akt and other proteins was down-regulated, indicating that VB inhibited tumor growth and metastasis through exosome-mediated miR-7-5p transmission (Wang et al., 2024b). |

(Continued)

| Nanomaterial | Monomer | Experimental model | Changes in tumor cells and tumor size | Particle size | Zeta potential | Stability | Drug loading ratio (DL %) = or drug loading capacity (LC%) | Encapsulation efficiency | Efficacy | Antitumor Mechanism |
|---|---|---|---|---|---|---|---|---|---|---|
| | Resveratrol | Huh7 cell line and nude mouse subcutaneous xenograft model | Inhibits tumor cell proliferation, migration, and epithelial-mesenchymal transition (EMT), significantly inhibiting tumor volume and weight | 30–150 nm | | | | | | Resveratrol inhibits exosome secretion in Huh7 cells by downregulating Rab27a, and inhibits the proliferation, migration, and epithelial-mesenchymal transition (EMT) of tumor cells through exosome delivery of lncRNA SNHG29. In addition, resveratrol-induced exosomes further inhibit tumor progression by inhibiting autophagy and the Wnt/β-catenin signaling pathway. The signaling pathway proteins involved include the autophagy markers Beclin1, p62, and LC3, as well as GSK-3β, β-catenin, and c-Myc in the Wnt/β-catenin pathway (*Negri et al., 2020*). |
| | Steroidal glycoalkaloids from *Solanum lyratum* | A549 Lung cancer cells and human umbilical vein endothelial cells (HUVECs) | Inhibition of the transformation of HUVECs into tumor-associated endothelial cells (Td-ECs) | | | | | | | *In vitro* experiments verified the anti-angiogenic activity of these compounds against tumor cells. Exosomes secreted by A549 cells promote the transformation of HUVECs into tumor-associated endothelial cells (Td-ECs), which is significantly inhibited by steroidal glycosalkaloids such as TSGS and SA1. These compounds may interfere with lipid raft function by aggregating cell membrane cholesterol, thereby inhibiting exosome formation and function, and ultimately inhibiting tumor angiogenesis. Signaling pathway proteins involved include VEGFR2 and its downstream signaling molecules (*Liu et al., 2021*). |
| Micelles | Baicalin | Hep G2 and Hela cell | Exhibits higher cytotoxicity in A549 cells | 15.60 nm | 5.26 mV | After 48 h of incubation in PBS, there was no significant change in particle size | LC% = 16.94%, | 90.67%. | It is not easily recognized and captured by the endothelial reticular system, can exist stably in the blood for a long time, can achieve passive targeting of tumor sites through EPR effect, can promote intracellular drug accumulation and anti-tumor effects, and can achieve effective drug release at tumor sites through the introduction of stimulus-responsive prodrugs, such as pH-responsive micelles. | Biotin-modified micelles bind to the biotin receptor (SMVT) on the surface of tumor cells, enter cells through receptor-mediated endocytosis, release paclitaxel (PTX) to exert cytotoxicity, and non-cancer cells (HEK293 cells, biotin receptor negative) have low micelle uptake (*Wang et al., 2020a*). |

| Nanomaterial | Monomer | Experimental model | Changes in tumor cells and tumor size | Particle size | Zeta potential | Stability | Drug loading ratio (DL %) = or drug loading capacity (LC%) | Encapsulation efficiency | Efficacy | Antitumor Mechanism |
|---|---|---|---|---|---|---|---|---|---|---|
| | Shikonin | CT26 cells (mouse colon cancer cells), H2 tumor-bearing mice | Tumor volume was significantly reduced, and the tumor inhibition rate reached 62.9% | 172.1 nm | −20.3 mV | Leave at room temperature for 7 days | LC% = 9.4 ± 0.6% | 82.4 ± 4.2% | | SK/Siidol-HMS can trigger immunogenic cell death (ICD), promote CRT exposure, increase the proportion of DCs maturation in draining lymph nodes up to 46.9 ± 7.3%, increase the infiltration of CD8 T cells within the tumor, increase intratumoral γ-interferon-γ (IFN-γ) and tumor necrosis factor-α α, TNF-α), reduce the expression level of IDO-1 and the ratio of kynurenine to tryptophan, reduce the frequency of intratumoral infiltrating regulatory T cells (Tregs), and increase antitumor effects compared with free monomers (Tong et al., 2024). |
| | Emodin | 4T1 breast cancer cells | Cell proliferation is inhibited and tumor volume decreases | 37.2 ± 1.7 nm, | 0.64 ± 0.3 mV | Store at 4 °C for 30 days | LC% = 13.4 ± 0.4% | 77.2 ± 2.7% | | Polymers induce apoptosis by depleting GSH and increasing ROS levels, activating the intracellular oxidative stress response. In addition, the structure of the polymer is disrupted in the tumor microenvironment, releasing drugs and enhancing the cellular uptake and accumulation of drugs, further improving the anti-tumor effect (Du et al., 2020). |
| | Camptothecin | SKOV3 ovarian cancer cells | Proliferation is significantly inhibited | 121.6 nm | −23.2 ± 0.5 mV | It exhibits good stability in simulated blood circulation (pH 7.4) and tumor cell microenvironment (pH 5.0). | LC% = 12.927 ± 0.880% | 85.097 ± 2.936% | | Enzyme responsiveness is achieved by cathepsin B-sensitive ALAL peptide, and a positively charged secondary micelle (9-NC/TPC) is formed after cleavage, which promotes lysosomal escape and enters the nucleus with the help of TAT peptide, inhibits topoisomerase I, and induces apoptosis of tumor cells. The micelles increase drug loading through π-π stacking to ensure stable drug delivery, while using the acidic environment and enzyme activity in tumor cells to achieve targeted release, enhancing the cellular uptake and intranuclear accumulation of drugs (Zheng et al., 2020). |

(Continued)

| Nanomaterial | Monomer | Experimental model | Changes in tumor cells and tumor size | Particle size | Zeta potential | Stability | Drug loading ratio (DL %) = or drug loading capacity (LC%) | Encapsulation efficiency | Efficacy | Antitumor Mechanism |
|---|---|---|---|---|---|---|---|---|---|---|
| | Andrographolide | CT26 colon cancer cells | Improves toxicity to tumor cells | 84.68 ± 2.08 nm, | −12.49 ± 2.28 mV | Store at room temperature for 7 days | DL% = 6.84 ± 0.01% | 91.15 ± 0.05% | | BSP-VES micelles significantly improved the anti-tumor effect of AG by enhancing cell uptake and tumor targeting. In the experiment, the proliferation of CT26 colon cancer cells was significantly inhibited, and the tumor volume was significantly reduced in the *in vivo* experiment. In addition, BSP-VES micelles improved their toxicity to tumor cells by enhancing the intracellular accumulation of AG while reducing damage to normal cells (*Zhao et al., 2012*). |
| | Emodin | | Apoptosis is initiated, the tumor volume is reduced by 70%, and it disappears completely after 2 weeks | 100 nm | | Medium with deionized water and 10% fetal bovine serum for 7 days | LC% = 73.8 ± 2.8% | | | Iron oxide nanocubes produce local high temperature (42–45 °C) under alternating magnetic field, triggering the upregulation of heat shock protein (HSP) expression, disrupting tumor cell homeostasis and activating the mitochondrial apoptosis pathway, high temperature induces increased cell membrane fluidity, promotes the flow state of the PHEP chain in the micelle core, accelerates the release of the hydrophobic drug emodin, which blocks DNA replication by inhibiting Topoisomerase II activity, and activates the MAPK pathway to induce cell cycle G2/M phase arrest, Magnetic targeting enhances drug accumulation at tumor sites, and local high concentration of emodin further down-regulates the expression of anti-apoptotic protein by inhibiting the NF-κB signaling pathway, and synergizes hyperthermia to induce programmed cell death (*Song et al., 2020*). |

| Nanomaterial | Monomer | Experimental model | Changes in tumor cells and tumor size | Particle size | Zeta potential | Stability | Drug loading ratio (DL %) = or drug loading capacity (LC%) | Encapsulation efficiency | Efficacy | Antitumor Mechanism |
|---|---|---|---|---|---|---|---|---|---|---|
| | Shikonin | MDA-MB-231 (Luc1) cell | Significantly inhibits the volume and weight of TNBC tumors | <70 nm | | It is well stable at pH 7.4 with 50% fetal bovine serum (FBS). | LC% > 50% | | | SK inhibits mitochondrial biosynthesis by binding to and inhibiting mitochondrial polymerase γ (POLG), which in turn downregulates the expression of PGC-1α (peroxisome proliferator-activating receptor γ coactivator 1α), inhibiting mitochondrial DNA (mtDNA) levels. This mechanism leads to a decrease in the proliferation and metabolic capacity of tumor cells, ultimately inhibiting tumor growth and metastasis. In the experiment, the expression of POL and PGC-1α in SK-treated TNBC cells and tumor tissues was significantly reduced, and the number of mitochondria and ATP levels in tumor cells were also significantly reduced. *In vivo* experiments, SK significantly inhibited the volume and weight of TNBC tumors and reduced the number and area of lung metastases (*Su et al., 2017*). |
| Carbon nanotubes | Curcumin | 4T1 breast cancer cell line | Reduced cell viability | 8.5 nm, | −14 eV | | DL% = 27.14%, | | With a hollow physical structure and easy to modify chemical properties, it can be developed for the delivery of therapeutic drug carriers, enhance the pharmacological activity of traditional TCM monomers, reduce drug side effects, and increase drug accumulation in tumor sites. | The drug release rate of CNT-Pt-CUR in the acidic environment (pH 4.7) was significantly higher than that in the physiological environment (pH 7.4), and it showed good pH sensitivity. In addition, combined with X-ray radiation, CNT-PT-CUR significantly enhanced the killing effect on tumor cells, and it was speculated that it could enhance the radiotherapy effect by increasing the production of reactive oxygen species (ROS), inducing DNA damage and cell cycle arrest (*Wang et al., 2020c*). |
| | Plumbagin | Bel-7402 tumor cells HepG-2 for liver cancer and HCT-116 for colon cancer | Significantly reduces the survival rate of tumor cells and induces apoptosis | 300 nm | | Good stability was shown in water, PBS, and complete media | | | | After PLB-PEG-SWNTs are internalized into cells, the amide and ester bonds that connect drug molecules and SWNTs are hydrolyzed and broken, and the drug molecules are directly released to directly act on the G2/M phase control points of tumor cells and initiate the apoptosis pathway, thereby affecting cell division and proliferation (*Li et al., 2021*). |

(Continued)

## Table 2 (continued)

| Nanomaterial | Monomer | Experimental model | Changes in tumor cells and tumor size | Particle size | Zeta potential | Stability | Drug loading ratio (DL%) = or drug loading capacity (LC%) | Encapsulation efficiency | Efficacy | Antitumor Mechanism |
|---|---|---|---|---|---|---|---|---|---|---|
| | Ginsenoside RG3 | TNBC cell | Promotes apoptosis of tumor cells, reducing tumor volume by approximately 65% | | | | | | | Rg3-CNT down-regulates the expression of PD-L1 protein on the surface of tumor cells, blocks its immunosuppressive binding to PD-1 on the surface of T cells, relieves immune escape, and inhibits the transcriptional expression of PD-L1 by inhibiting the mRNA and protein levels of the epiregulatory factor BRD4. Rg3-CNT reduced tumor volume by about 65%, delayed growth by about 18 days, and decreased PD-L1 expression in tumor tissues by 40–50% (*Liang et al., 2023*). |
| | Paclitaxel | L929 mouse embryonic fibroblasts and Hela cell lines | Significantly inhibited the growth of Hela cells | 65–105 nm | | 74.4% release rate within 96 h | LC% = 58–77%, | 29–36.5% | | CNTs-g-PMAA enables targeted therapy of tumor cells through the rapid release of anti-cancer drugs (e. g., paclitaxel, PTX) in an acidic environment. This mechanism exploits the acidic microenvironment of tumor cells (pH 5.8–6.8) in contrast to the normal physiological environment (pH 7.4), allowing for precise drug release, enhancing anti-cancer effects and reducing toxicity to normal cells. PTX-loaded CNTs-g-PMAA had a better inhibitory effect on tumor cells than free PTX in an acidic environment, and had lower toxicity to normal cells (*Shao et al., 2015*). |
| Dendrimers | Honokiol | Huh-7 and HepG-2 hepatoma cells | Inhibit tumor cell proliferation and promote tumor cell apoptosis | 119.9 to 199.5 nm | −10.9 mV to −22.3 | It exhibits good stability in both water and ethanol | DL% = 1.4 to 5.37% | 5.37% | The surface functional groups can be covalently coupled to the monomer of traditional TCM, and the drug loading is controllable. It has active targeting, and the end can modify the target molecule (such as RGD peptide) to improve tumor specificity. The positively charged surface promotes cell uptake and has high transmembrane efficiency. | The ability to inhibit MMP-9 is conferred by modification of latent zinc-bound branched linkers and terminals. The synthetic Third - Generation Carboxylic Acid - Terminated Dendrimer (TPG 3-Terminated Dendrimer (TPG 3-OH) 10) and hydrazide analogue ((TPG 3-NH₂) 12) were safer against normal lung fibroblasts than Honokiol. (TPG3-NH₂)12 has better inhibitory activity than (TPG3-OH)10 for MMP-9 and MMP-2 (*Mohd Nurazzi et al., 2021*). |

| Nanomaterial | Monomer | Experimental model | Changes in tumor cells and tumor size | Particle size | Zeta potential | Stability | Drug loading ratio (DL%) = or drug loading capacity (LC%) | Encapsulation efficiency | Efficacy | Antitumor Mechanism |
|---|---|---|---|---|---|---|---|---|---|---|
| | Curcumin | Glioblastoma cells | Significantly reduces the activity of tumor cells | 4 nm | −32.4 mV ± 0.4 | 10.1 min | DL% = 10% | 85.097 ± 2.936% | | G4 90/10-Cys dendritic macromolecule can effectively encapsulate curcumin and significantly reduce the activity of human (U87), murine (GL261), and rat (F98) glioblastoma cells, with less effect on normal cells (e.g., HEK 293, mouse and rat bone marrow mesenchymal stem cells). Encapsulated curcumin significantly improves anti-tumor efficacy by increasing its bioavailability (*Mohammadi et al., 2024*). |
| | Conjugated podophyllo toxin | Mouse model of hepatocellular carcinoma (HCC). | Inhibits liver fibrosis and hepatic stellate cell activation | | | | DL% = 9:1 | | | Reduces interleukin-6 (IL-6) levels and inhibits its induced DNA damage and tumor growth; 2) inhibition of the expression of nuclear factor kappa B (NF-κB) and blocking the key link between inflammation and cancer; 3) Reduce the expression of transforming growth factor-β (TGF-β) and α-smooth muscle actin (α-SMA), and inhibit liver fibrosis and hepatic stellate cell activation. Together, these effects inhibit the progression of HCC (*Zhong, 2011*) |
| | Resveratrol | Lung cancer A549 cells | Inhibits cell proliferation | 251.4 ± 15.7 nm | | Store at room temperature for a long time | LC% = 18.4 ± 0.9% | 75.1 ± 2.2% | | Through ester bond bonding, a large number of drugs are released after complete hydrolysis of ester bonds catalyzed by enzymes in esterase-rich tumor cells, enhancing the anticancer activity of drugs (*Zhou et al., 2020*) |
| Nanogels | Emodin | HepG2 hepatocellular carcinoma cell line and a nude mouse subcutaneous xenograft model | Induces apoptosis and reduces tumor volume and weight | 80 nm | −24.13 mV at pH 7.4 and 1.21 mV at pH 6.5 | Store at 25 °C for 14 days | DL% = 100% | | The three-dimensional cross-linked network structure is suitable for loading water-soluble TCM monomers, intelligent response release such as pH/temperature response, tumor microenvironment triggers gel swelling or contraction, release of drugs, has good biocompatibility, degradation products are non-toxic, and the surface can be modified to target ligands (such as folic acid) or immune checkpoint inhibitors (PD-L1 antibodies) | By targeting mitochondria, Rhein–DOX nanogel significantly increases intracellular reactive oxygen species (ROS) levels, decreases mitochondrial membrane potential (ΔΨm), and induces apoptosis. These changes suggest that nanogels exert anti-tumor effects by enhancing ROS-mediated mitochondrial damage and inducing apoptosis. Signaling pathway proteins involved include proteins related to mitochondrial function (e.g., JC-1) and apoptosis-related proteins (e.g., Annexin V) (*Singh & Kesharwani, 2021*) |

(Continued)

| Nanomaterial | Monomer | Experimental model | Changes in tumor cells and tumor size | Particle size | Zeta potential | Stability | Drug loading ratio (DL %) = or drug loading capacity (LC%) | Encapsulation efficiency | Efficacy | Antitumor Mechanism |
|---|---|---|---|---|---|---|---|---|---|---|
| MOFs contain ferrous metal-organic frameworks | Triptonide | | Activates dendritic cell maturation (DCs) maturation and T cell infiltration | 100 nm | −17 mV | Store at 4 °C 30 days | DL% = 36.2% | 81.4% | The porous structure can efficiently load hydrophobic TCM monomers and encapsulate chemotherapy drugs at the same time. $Fe^{2+}/Fe^{3+}$ produces reactive oxygen species (ROS) through the Fenton reaction, which directly kills tumor cells, synergizes with the anti-tumor mechanism of traditional TCM monomers, and has imaging functions. | TPL inhibits Nrf2 expression and interferes with *de novo* glutathione (GSH) synthesis, increasing cell sensitivity to ferroptosis; At the same time, $Fe^{3+}$ increased the intracellular ROS level through the Fenton reaction after entering the cell, and the two synergistically amplified the production of ROS and induced ferroptosis, which was manifested by down-regulation of Nrf2 and GPX4 expression and decreased GSH level. On the other hand, TPL can induce GSDME-dependent pyroptosis, and $Fe^{3+}$ enhances this process by activating caspase-3, causing it to cleave GSDME to produce N fragments, forming pores in the cell membrane, causing cell swelling, membrane rupture, and the release of IL-1β and LDH, *etc.*, to trigger pyroptosis (*Sharma et al., 2017*) |
| Self-assembling nano particles | Ursolic acid | 3T3 cells LNCaP cells | Targeting CAF cells and regulating the tumor microenvironment | 195.13 ± 8.06 nm, | −29.07 ± 0.55 mV | Store at 4 °C for 30 days | LC% = 12.93%, | 92.80% | The hydrophobic nucleus is loaded with fat-soluble drugs, the hydrophilic shell enhances stability, the photosensitive or redox-sensitive bonds are designed, the targeted modification is convenient, the surface is easy to couple the targeted molecule, and the self-assembly process is simple, which is suitable for industrial scale-up. | Anisamide-modified ursolic acid nanoparticles can significantly down-regulate the expression of NRG1 in CAF cells, thereby reducing the phosphorylation levels of HER3 and AKT in LNCaP cells, thereby alleviating enzalutamide resistance. This mechanism suggests that the nanoparticles modulate signaling pathways in the tumor microenvironment by targeting CAF cells and enhance the sensitivity of prostate cancer cells to enzalutamide (*Khalil et al., 2021*) |

albumin nanotechnology can solve the patient's allergy problem and improve the anti-tumor effect. The reason why the efficacy of albumin paclitaxel can be improved is that the nanocarrier can quickly deliver the drug to the cancer tissue and retain it for a longer time. In the treatment of breast cancer, conventional paclitaxel treatment can only achieve a remission rate of 19%, while albumin paclitaxel can increase it to 33% (*Gupta & Gupta, 2022*). In a clinical trial (CA201) for the first-line treatment of advanced breast cancer conducted in China, the objective response rate after albumin-paclitaxel treatment reached 56%, while the control group using conventional chemotherapy drugs could only reach 27% (*Guo et al., 2019*). In addition, albumin-paclitaxel also greatly reduced its toxicity to neutrophils, and the incidence of severe neutropenia after treatment was only half that of paclitaxel. It also greatly shortened the time required for injection, and the injection can be completed in half an hour. These clinical experiments have strongly confirmed the effectiveness and potential of nanomaterial-loaded TCM monomers in anti-tumor treatment, providing a solid foundation for its further clinical application.

# POTENTIAL OF NANO-TCM MONOMERS IN COMBINATION WITH OTHER THERAPIES

## Potential of nano-TCM monomers in combination with radiotherapy

The unique size and structure of nanomaterials enable them to more effectively deliver TCM monomers to tumor sites (*Zhang et al., 2024*). Radiotherapy is one of the important means of tumor treatment, but the radiotherapy resistance of tumor cells and the damage of radiotherapy to normal tissues limit its efficacy. The combination of nanomaterial-loaded TCM monomer delivery system and radiotherapy shows great potential to overcome these problems. Nanomaterials have unique size, shape and surface properties, which can change the interaction between radiotherapy rays and tumor tissues. Nanomaterials can be used as radiotherapy sensitizers to enhance the absorption and scattering of rays, increase local energy deposition, and thus increase radiation damage to tumor cells. At the same time, its size advantage enables it to passively or actively target tumor tissues, increase the enrichment of drugs in tumor sites, and achieve synergistic enhancement of radiotherapy and drug therapy (*Qian et al., 2022*). In terms of tumor microenvironment regulation, nanomaterial-loaded TCM monomer delivery system can effectively improve the adverse microenvironment faced by radiotherapy. The hypoxic state of the tumor microenvironment is an important factor leading to radiotherapy resistance. Some TCM monomers can promote the normalization of tumor blood vessels and increase oxygen supply. Nanomaterials can accurately deliver these TCM monomers to the tumor area, improve the hypoxic microenvironment, and increase radiotherapy sensitivity. In addition, there are immunosuppressive cells and factors in the tumor microenvironment. The TCM monomers delivered by nanomaterials can regulate the immune microenvironment, activate immune cells, inhibit the function of immunosuppressive cells, and enhance the body's immune clearance ability of tumor cells after radiotherapy, forming a combined effect of radiotherapy and immunotherapy. For tumor cells, the combination of nanomaterial-loaded TCM monomer delivery system and

radiotherapy has a multi-faceted effect on their biological behavior. Radiotherapy induces DNA damage in tumor cells, and TCM monomers can affect the DNA damage repair mechanism of tumor cells through various pathways (*Deng et al., 2023b*). At the same time, nanomaterials can accurately deliver TCM monomers into tumor cells, enhance their regulatory effect on the cell cycle, and make tumor cells more stagnant in the cell cycle phase sensitive to radiotherapy, thereby improving the efficacy of radiotherapy.

## Potential of combining nano-TCM monomers with immunotherapy

Immunotherapy has become an important breakthrough in the field of tumor treatment, but it still faces many challenges, such as immune escape and insufficient immune activation. The combination of nano TCM monomer delivery system and immunotherapy shows great potential to overcome these problems. Nanomaterials can effectively improve the pharmacokinetic properties of TCM monomers with their unique size, shape and surface properties, and achieve precise targeted delivery to the tumor microenvironment and immune cells. From the perspective of antigen presentation, nanomaterials can encapsulate or adsorb TCM monomers and deliver them to antigen-presenting cells. TCM monomers may promote the maturation of dendritic cells and enhance their ability to take up, process and present tumor antigens, thereby activating naive T cells and initiating anti-tumor immune responses (*Shen et al., 2024*). There are immunosuppressive cells and factors in the tumor microenvironment, such as regulatory T cells (Tregs), myeloid-derived suppressor cells (MDSCs) and immune checkpoint molecules, which hinder the effectiveness of immunotherapy. Nano TCM monomer delivery system can regulate this unfavorable environment. TCM monomers may enhance the activity, proliferation and cytotoxicity of immune effector cells by regulating intracellular signaling pathways, thereby significantly improving their ability to kill tumor cells (*Miao et al., 2023*). *Wang et al. (2020b)* constructed an enzyme-sensitive AP (Angelica Polysaccharide)-PP-DOX (doxorubicin) tumor-targeted nanodrug delivery system (PP MMP2-cleavable peptide). MMP2 can cleave AP-PP-DOX (Angelica 1 doxorubicin) to release the AP part (AP-GPLG). AP-GPLG does not affect the immunomodulatory activity of AP. It increases the proliferation of mouse spleen cells and regulates Th1/Th2 cytokines (upregulating IL-2 and downregulating IL-10), which is expected to restore the immune balance of the tumor microenvironment and fight tumors. The combination of TCM-loaded monomer delivery system and immunotherapy has opened up a new path to overcome the difficulties of immunotherapy and improve the effect of tumor treatment through the synergistic effects of multiple mechanisms such as optimizing antigen presentation, reshaping the tumor immune microenvironment, and activating immune effector cells. It is expected to bring better treatment options for cancer patients.

## SUMMARY AND OUTLOOK

Tumors pose a serious threat to global public health, and the incidence and mortality rates have increased significantly with the aging of the population. Chemotherapy is the most commonly used method for treating tumors. However, due to the small metabolic differences between tumors and normal cells, chemotherapy has poor targeting, is prone to

cause adverse reactions such as bone marrow suppression and liver and kidney dysfunction, and is expensive, which greatly limits the therapeutic effect. TCM monomers have the advantages of multi-target effects, low toxicity and synergistic drug delivery in tumor treatment. However, most active ingredients of TCM have problems such as poor solubility and stability, poor tissue permeability, rapid clearance in the body and short half-life, which leads to the inability to accumulate in the target tissue and poor drug delivery effect in tumors. This study reviews the therapeutic effects of using TCM monomer nano-delivery systems in anti-tumor treatment, providing ideas for further exploration of anti-tumor mechanisms. The article mainly mentions nano-carriers such as lipid nanomaterials, extracellular vesicles, polymer micelles, carbon nanotubes, and dendritic macromolecules, which have the functions of increasing the bioavailability of TCM monomers, achieving targeted delivery, controlling drug release, and inhibiting multidrug resistance of tumor cells. For example, lipid nanomaterials can improve the stability and biocompatibility of TCM monomers; exosomes, as natural drug delivery systems, can use their immunomodulatory and tumor microcirculation remodeling properties for anti-tumor treatment; polymer micelles can achieve passive targeting and have good water solubility; carbon nanotubes can enhance the pharmacological activity of TCM monomers; dendritic macromolecules can improve drug solubility and bioavailability, *etc.*On the one hand, TCM monomers can inhibit tumor cell proliferation and metastasis through various pathways, such as promoting tumor cell apoptosis, inducing cell autophagy, *etc.* On the other hand, the delivery system of TCM monomers loaded with nanomaterials plays an important role in regulating the TME. From the perspective of nanomaterial characteristics, their size, shape, and surface charge determine the pathway of action in the TME, and surface modification can regulate the interaction with the various components of the TME; at the cellular level, the system can regulate the function of immune cells in the TME, inhibit the activity of immunosuppressive cells, regulate the function of tumor-associated fibroblasts, and affect tumor angiogenesis, thereby inhibiting tumor growth and metastasis.

At present, the development of TCM nanoformulations faces many challenges. Although nanomaterials loaded with TCM monomers have shown significant advantages in anti-tumor therapy and brought new hope for overcoming tumor problems, their many shortcomings have also brought a series of challenges. The potential toxicity of nanomaterials loaded with TCM monomers is a key issue that needs to be addressed. The small size of nanocarriers enables them to easily penetrate biological membranes, interact with biomacromolecules in cells, interfere with the normal physiological functions of cells, and induce potential toxic side effects. In addition, after the nanocarriers combine with TCM monomers, the complexes formed may produce new toxic mechanisms, and the current research on the mode of action and long-term effects of these toxicities is still in the exploratory stage, which greatly limits the wide application of nanomaterials loaded with TCM monomers in clinical treatment. Future research needs to establish standardized toxicity assessment methods, and at the same time, more deeply evaluate the biocompatibility and potential toxicity of nanoparticles, and reduce the toxicity of nanoparticles by designing safer nanomaterials, such as using biodegradable materials or

optimizing surface modifications. Long-term stability is also an important factor restricting its development. In the complex and changeable physiological environment in the body, nanomaterials loaded with TCM monomers are susceptible to multiple factors. The pH differences between different tissues and organs in the body, enzymes, and oxidative stress environments may destroy the structure of nanocarriers, leading to premature leakage or loss of activity of TCM monomers. In addition, the modified functional groups on the surface of nanoparticles may fall off over time, resulting in a decrease in their targeting ability and drug release characteristics. This not only fails to ensure the integrity of the drug before it reaches the target site, but also may reduce its targeted therapeutic effect on tumor cells, seriously affecting the therapeutic effect on tumors. It is possible to consider using biocompatible materials to enhance the stability of nanoparticles, so as to avoid the damage caused by the long-term accumulation of nanoparticles in the body, and to avoid aggregation, degradation, or changes in chemical properties during the storage and use of nanoparticles. The complexity of the amplification process is also an obstacle for nano TCM monomers to move from the laboratory to the clinic. The transition from small-scale preparation in the laboratory to large-scale industrial production requires substantial adjustments and optimization of the preparation process. However, key parameters such as the particle size distribution, morphological structure, and drug loading of nanocarriers cannot be kept consistent during the amplification process. In addition, large-scale production also needs to comprehensively consider factors such as cost-effectiveness, quality control, and environmental impact. Under the premise of ensuring product quality, achieving efficient and low-cost large-scale production is a difficult problem that needs to be solved in the future development of nano TCM monomers. In the future, efficient nano-drug amplification production processes can be studied and developed to reduce costs and improve production efficiency. For example, continuous flow synthesis technology or microfluidics technology can be used to achieve high-throughput production of nanoparticles, while also preparing products of the same quality level. The development of nanotechnology requires close cooperation among multiple disciplines such as chemistry, materials science, biology, and medicine. Through interdisciplinary research, the challenges of nanoparticles in biomedical applications can be better addressed and their clinical transformation can be promoted. In short, although nanoparticles have great potential in the biomedical field, their limitations cannot be ignored. Nano-TCM monomers bring opportunities and challenges to tumor treatment. Under the support and guidance of modern TCM theory, these challenges can be overcome through in-depth research and technological innovation to promote more complete, safe, and effective development of tumor treatment, and make important contributions to the international promotion, application, innovation, and green development of TCM.

### Funding

This review was supported by the Postdoctoral Scientific Research Start-up Fund of Heilongjiang (LBH-Q21158), Heilongjiang Bayi Agricultural University Experimental Demonstration Base Project: Development and Application of Corn Straw Biofeed for Improving the Growth Performance of Poultry and Heilongjiang Bayi Agricultural University Graduate Innovative Research Project (YJSCX2023-Y69). The funders had no role in study design, data collection and analysis, decision to publish, or preparation of the manuscript.

### Grant Disclosures

The following grant information was disclosed by the authors:
Postdoctoral Scientific Research Start-up Fund of Heilongjiang: LBH-Q21158.
Heilongjiang Bayi Agricultural University Experimental Demonstration Base Project: Development and Application of Corn Straw Biofeed for Improving the Growth Performance of Poultry and Heilongjiang Bayi Agricultural University: YJSCX2023-Y69.

### Competing Interests

Shuang Zhang is an employee of the Comprehensive Service Center.

### Author Contributions

- Bocui Song conceived and designed the experiments, authored or reviewed drafts of the article, and approved the final draft.
- Li Shuang conceived and designed the experiments, authored or reviewed drafts of the article, and approved the final draft.
- Shuang Zhang conceived and designed the experiments, prepared figures and/or tables, and approved the final draft.
- Chunyu Tong analyzed the data, authored or reviewed drafts of the article, and approved the final draft.
- Qian Chen performed the experiments, prepared figures and/or tables, and approved the final draft.
- Yuqi Li performed the experiments, prepared figures and/or tables, and approved the final draft.
- Meihan Hao performed the experiments, prepared figures and/or tables, and approved the final draft.
- Wenqi Niu analyzed the data, authored or reviewed drafts of the article, and approved the final draft.
- Cheng-Hao Jin analyzed the data, authored or reviewed drafts of the article, and approved the final draft.

### Data Availability

This is a literature review.

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
