# Peer review of "Research progress of nano drug delivery systems in the anti-tumor treatment of traditional Chinese medicine monomers"

_PeerJ, doi:10.7717/peerj.19332_

## Round 0.1 · original submission · Major Revisions

Dear authors,

Thank you for your submission. At the moment, significant revisions are required. Please, refer to the reviewers comments, and present a point-by-point rebuttal.

Reviewer 1 ·

Basic reporting

1. The article has been well written, with a clear structure and easy-to-follow flow. The author effectively presents the topic, providing sufficient explanations for each point discussed. However, some references used in the article are somewhat outdated (more than 5 years old) and need to be updated. Updating these references will ensure that the article encompasses the latest developments in the field, maintaining its relevance.


2. The author has written the article in accordance with the scope of the targeted journal. The article presents a relevant topic and can make a significant contribution to the developing field. The topic discussed is quite interesting and has the potential to attract the readers' attention, making it worthy of publication.



3. The introduction of this article has been well structured and provides a solid foundation for further discussion on the topic. The author successfully explains the importance of this topic and connects it to current developments, allowing readers to easily follow the arguments presented in the article.

Experimental design

1. This review is well-aligned with the scope of the journal. The topic is relevant and falls within the journal's focus, making it suitable for publication.

2. In general, the methodology is described in sufficient detail. However, the author does not mention the total number of data points obtained or provide information regarding inclusion or exclusion criteria, nor do they specify the time frame limitations for the data considered. Including these details would enhance the transparency and rigor of the study.

3. The review is generally well-organized, with a clear structure that facilitates ease of understanding. The logical flow of the discussion and the connection between different sections make it easy for the reader to follow the arguments presented.

Validity of the findings

1. Although reviews on this topic are relatively abundant, there is a lack of reviews specifically focusing on Traditional Chinese Medicine (TCM) monomers. The author is encouraged to highlight a more distinct perspective, clearly emphasizing the novelty of the review. It is important to clearly articulate the unique contribution of this work and the urgency of addressing the topic, demonstrating its significance in the context of current research and applications.


2. The conclusion remains quite general and would benefit from further refinement. It should be sharpened to better encapsulate the key findings, the implications of the review, and potential future directions in the field. Providing a more focused and precise conclusion would significantly enhance the impact of the article.

Additional comments

1. The data presented in the tables should be expanded to provide more informative insights, especially regarding the outcomes of current research, such as cytotoxicity or tumor size reduction. Including these quantitative indicators will allow for a better comparison of which nanoparticle carriers show the most promise for delivering Traditional Chinese Medicine (TCM) monomers. This approach would provide readers with a clearer understanding of the practical applications and effectiveness of different nanoparticle formulations.


2. It is recommended to enhance Table 2 by adding details on the types of nanoparticles utilized in the studies, along with their respective characterizations. This would offer a more comprehensive overview of the materials employed and shed light on the techniques and methods used for nanoparticle characterization. Including this information will enhance the reader’s understanding of the materials and methodologies in the research, contributing to a more thorough analysis.


3. In addition to the typical considerations of bioavailability enhancement, controlled release, and targeting, it is crucial to include the concept of Multi-Drug Resistance (MDR) in the discussion of nanoparticle performance. Nanoparticles are known to reduce MDR, which is a significant advantage in the delivery of therapeutic agents. Incorporating this aspect would enrich the review and provide a more comprehensive perspective on the advantages of nanoparticles in drug delivery.


4. It is also essential to address the limitations of nanoparticles in the Summary Outlook section. While nanoparticles offer many advantages, they come with certain challenges, such as potential toxicity, long-term stability issues, and the complexity of scale-up processes. Acknowledging these limitations will provide a balanced view of nanoparticle-based systems and guide future research directions.


5. Ensure that any images or figures included in the manuscript are properly checked for copyright. It is essential to verify that all images used are either original or appropriately licensed to avoid copyright infringement.

Reviewer 2 ·

Basic reporting

- the limitations of each nanomaterial (e.g., toxicity, stability, scalability) are not thoroughly discussed. A deeper critique of the challenges and potential risks associated with these systems is necessary.

- it does not adequately address the potential toxicity of nanomaterials, especially in long-term use

- Including more clinical evidence would strengthen the argument for the practical application of these systems

- the molecular pathways and interactions are not explored in sufficient depth.

- The paper does not compare the efficacy of different nano-drug delivery systems (e.g., liposomes vs. exosomes vs. carbon nanotubes) in a systematic way.

- The paper does not address how nano-drug delivery systems might overcome drug resistance in tumors, which is a critical issue in cancer treatment. Including a section on this topic would add significant value.


- While the TME is mentioned, the discussion is superficial. A deeper exploration of how nano-drug delivery systems interact with and modulate the TME would be valuable.

- The paper does not explore the potential of combining nano-drug delivery systems with other therapies (e.g., immunotherapy, radiotherapy).

- The abstract is too brief and does not adequately summarize the key findings or implications of the review.

- The figures are informative but lack detailed captions.

- The conclusion is too brief and does not adequately summarize the key findings or future directions.

Experimental design

REFER TO 1. Basic reporting

Validity of the findings

REFER TO 1. Basic reporting

Additional comments

REFER TO 1. Basic reporting

Reviewer 3 ·

Basic reporting

No Comments

Experimental design

The review article was well researched and comprehensive. However, recent reviews on nanotechnology and TCM shall be included like https://www.frontiersin.org/journals/pharmacology/articles/10.3389/fphar.2024.1405252/full, https://doi.org/10.1007/978-981-16-3444-4_14.

Validity of the findings

No comments.

Annotated reviews are not available for download in order to protect the identity of reviewers who chose to remain anonymous.

---

## Round 0.2 · accepted · Accept

Dear authors,
i am now recommending your manuscript for publication. Thank your for your submission and collaboration. All the best!

Reviewer 2 ·

Basic reporting

Accept

Experimental design

Accept

Validity of the findings

Accept

Additional comments

Accept

Reviewer 3 ·

Basic reporting

The authors have made suggested changes and now Review is enriched.

Experimental design

Suggested changes have been satisfactorily addressed.

Validity of the findings

Suggested changes have been satisfactorily addressed.

Additional comments

No Comments